# Squalene emulsion-based vaccine adjuvants stimulate CD8 T cell, but not antibody responses, through a RIPK3-dependent pathway

Eui Ho Kim[1,2,3†], Matthew C Woodruff[1,2†], Lilit Grigoryan[4], Barbara Maier[5], Song Hee Lee[1,2], Pratyusha Mandal[1,6], Mario Cortese[4], Muktha S Natrajan[1], Rajesh Ravindran[1,2], Huailiang Ma[4], Miriam Merad[5], Alexander D Gitlin[7,8], Edward S Mocarski[1,6], Joshy Jacob[1,2,6], Bali Pulendran[4,8,9]*

[1]Emory Vaccine Center, Emory University, Atlanta, United States; [2]Yerkes National Primate Research Center, Emory University, Atlanta, United States; [3]Viral Immunology Laboratory, Institut Pasteur Korea, Seongnam, Republic of Korea; [4]Institute for Immunity, Transplantation and Infection, Stanford University School of Medicine, Stanford University, Stanford, United States; [5]Department of Oncological Sciences, Tisch Cancer Institute and the Immunology Institute, Icahn School of Medicine at Mount Sinai, New York City, United States; [6]Department of Microbiology and Immunology, Emory Vaccine Center, School of Medicine, Emory University, Atlanta, United States; [7]Department of Physiological Chemistry, Genentech, South San Francisco, United States; [8]Department of Pathology, Stanford University School of Medicine, Stanford University, Stanford, United States; [9]Department of Microbiology and Immunology, Stanford University School of Medicine, Stanford University, Stanford, United States

*For correspondence:
bpulend@stanford.edu

[†]These authors contributed equally to this work

Competing interests: The authors declare that no competing interests exist.

**Abstract** The squalene-based oil-in-water emulsion (SE) vaccine adjuvant MF59 has been administered to more than 100 million people in more than 30 countries, in both seasonal and pandemic influenza vaccines. Despite its wide use and efficacy, its mechanisms of action remain unclear. In this study we demonstrate that immunization of mice with MF59 or its mimetic AddaVax (AV) plus soluble antigen results in robust antigen-specific antibody and CD8 T cell responses in lymph nodes and non-lymphoid tissues. Immunization triggered rapid RIPK3-kinase dependent necroptosis in the lymph node which peaked at 6 hr, followed by a sequential wave of apoptosis. Immunization with alum plus antigen did not induce RIPK3-dependent signaling. RIPK3-dependent signaling induced by MF59 or AV was essential for cross-presentation of antigen to CD8 T cells by Batf3-dependent CD8+ DCs. Consistent with this, RIPK3 deficient or Batf3 deficient mice were impaired in their ability to mount adjuvant-enhanced CD8 T cell responses. However, CD8 T cell responses were unaffected in mice deficient in MLKL, a downstream mediator of necroptosis. Surprisingly, antibody responses were unaffected in RIPK3-kinase or Batf3 deficient mice. In contrast, antibody responses were impaired by in vivo administration of the pan-caspase inhibitor Z-VAD-FMK, but normal in caspase-1 deficient mice, suggesting a contribution from apoptotic caspases, in the induction of antibody responses. These results demonstrate that squalene emulsion-based vaccine adjuvants induce antigen-specific CD8 T cell and antibody responses, through RIPK3-dependent and-independent pathways, respectively.

## Introduction

The discovery of vaccine adjuvants which enhance the magnitude and durability of immune response to antigens has greatly facilitated the development of effective vaccines against diseases such as influenza, hepatitis B, malaria and shingles (*Coffman et al., 2010*; *Levitz and Golenbock, 2012*; *McKee and Marrack, 2017*; *O'Hagan et al., 2017*; *Pulendran and Ahmed, 2011*; *Reed et al., 2013*). Adjuvants comprise a variety of substances, ranging from emulsions such as aluminum salts (alum), to purified plant extracts such as QS21, to synthetic nanoparticles and small molecules that activate specific receptors in the innate immune system (*Del Giudice et al., 2018*; *O'Hagan et al., 2017*). The benchmark for adjuvant research has been alum which has been included as a safe and effective adjuvant in billions of doses of vaccines, administered to diverse populations throughout the world over the past 80 years. Alum has long remained the only adjuvant licensed for clinical use, but, after decades of slow progress, recent years have witnessed the licensure of several human vaccines containing novel adjuvants, such as the squalene emulsion-based adjuvants MF59 and AS03, the Toll-like receptor 4 (TLR4) agonist monophosphoryl lipid A (MPL) absorbed on alum (AS04), and combined immune stimulators such as QS21 and MPL (AS01) (*Didierlaurent et al., 2017*; *Garçon and Di Pasquale, 2017*; *MacLeod et al., 2011*). Many adjuvants can induce activation of the innate immune system, which programs the magnitude, quality and durability of the adaptive immune response (*Coffman et al., 2010*; *Levitz and Golenbock, 2012*; *O'Hagan et al., 2017*; *Pulendran and Ahmed, 2011*; *Reed et al., 2013*). Importantly, seminal advances in our understanding of innate immunity over the past two decades (*Beutler et al., 2006*; *Iwasaki and Medzhitov, 2015*; *Satoh and Akira, 2016*; *Temizoz et al., 2018*) have facilitated the evaluation of novel synthetic adjuvants that target innate immune receptors such as TLRs (*Hanson et al., 2015*; *Hou et al., 2011*; *Kasturi et al., 2011*; *Kurche et al., 2012*; *Lynn et al., 2015*; *MacLeod et al., 2011*; *Petitdemange et al., 2019*; *Reed et al., 2013*; *Yamamoto et al., 2019*).

Whilst there has been much progress in understanding the cellular and molecular mechanisms of action of synthetic adjuvants such as TLR ligands, substantial knowledge gaps exist in our understanding of the mechanisms of action of classic adjuvants such as alum and MF59. Although it has long been a prevailing view that alum was 'immunologically inert,' and mediated its adjuvant effects via a 'depot effect' of slow release of antigens, injection of alum is known to rapidly recruit various cells, including neutrophils which expel neutrophil extracellular traps (NETs) composed of chromatin (*Munks et al., 2010*; *Walls, 1977*). Moreover, DNA released in NETs has been reported to mediate the adjuvant activity of alum (*Marichal et al., 2011*; *McKee et al., 2013*), although a recent study suggests this may in part be due to contaminations in the DNA preparations (*Noges et al., 2016*). Furthermore, alum is known to rapidly activate NALP3 inflammasome (*Eisenbarth et al., 2008*; *Li et al., 2007*; *McKee et al., 2009*), although opinions vary about the relative importance of this in mediating its immunogenicity (*De Gregorio et al., 2008*; *Eisenbarth et al., 2008*; *Franchi and Núñez, 2008*; *Li et al., 2007*; *McKee et al., 2009*; *Schmitz et al., 2003*).

In the case of MF59, recent studies demonstrate that it induces a broader range of cytokines and chemokines than alum or CpG DNA, and more rapidly recruits CD11b$^+$ inflammatory cells to the site of injection (*Mosca et al., 2008*). Interestingly, while the adjuvant effects of MF59 appear to be independent of the NALP3 inflammasome, mice deficient in ASC – a protein necessary for inflammasome activation – are impaired in their ability to mount antibody responses to MF59 adjuvanted proteins, arguing for an NALP3-independent role for ASC in mediating MF59 adjuvanticity (*Ellebedy et al., 2011*). Furthermore, although MF59 does not activate TLRs in vitro, MyD88 deficient mice were impaired in their ability to mount bactericidal antibody responses to the MF59 adjuvanted rMenB vaccine (*Seubert et al., 2011*). In addition, it has been demonstrated that transient ATP release by muscle is necessary for MF59-induced immune responses (*Vono et al., 2013*). Despite these important observations, there is a paucity of understanding about the cellular and molecular mechanisms that mediate the adjuvant effects of MF59. In the present study, we demonstrate that MF59 and its SE mimetic AddaVax can induce CD8 T cell responses through RIPK3-dependent cell signaling in LN-resident macrophages. Further, we show that RIPK3 signaling in these macrophages is essential for Batf3$^+$ dendritic cell-dependent Ag cross-presentation. Surprisingly, while the RIPK3 pathway was critical for adjuvant-triggered CD8 T cell response, it was dispensable for antibody responses that instead relied on apoptosis and damage-associated molecular pattern (DAMP) signaling. These observations reveal new mechanistic insights about MF59 and demonstrate that cellular immunity

and humoral immunity can be differentially regulated by RIPK3-dependent and -independent pathways, respectively.

## Results

### MF59 and AddaVax induce robust antibody and CD8 T cell responses

We compared the adjuvant effects of two types of clinically licensed vaccine adjuvants, alum and SEs (MF59 or its mimetic AddaVax (AV)) in mice (*Figure 1A*). We immunized and boosted C57BL/6 (B6) mice with chicken ovalbumin (Ova) adjuvanted with either alum, AV or MF59 (*Figure 1—figure supplement 1A*), and assessed the antibody response. The titers of Ova-specific IgG1 were comparable between the alum and SE groups, while titers of IgG2b and IgG2c (representing $T_{H1}$ IgG response) were significantly elevated in SE groups (*Figure 1B*). Consistent with this observation, the frequency of IFN-γ-producing OVA-specific CD4 T ($T_{H1}$) cells was increased in AV group (*Figure 1—figure supplement 1B*). Moreover, the generation of follicular helper T cells ($T_{FH}$) was also enhanced in SE groups (*Figure 1C*), consistent with a higher frequency of germinal center (GC) B cells (*Figure 1D and E*). These data suggest that SE adjuvants elicit stronger IgG2b and IgG2c responses by potentiating helper $T_{H1}$ and $T_{FH}$ cell activity and GC formation.

In addition to antibody production, Ova-specific CD8 T cell response in the AV group was significantly higher than in the alum group at day 7 post-boost (*Figure 1F*). Moreover, AV-induced Ova-specific CD8 T cells displayed enhanced expression of granzyme B, a surrogate indicator of cytotoxic function (*Figure 1G*). Notably, Ova-specific CD8 T cells in the SE groups could be found systemically in non-lymphoid organs, but did not accumulate in the gut (*Figure 1H,I and J*). The role of this cytotoxic T cell population was tested with a tumor challenge model using B16 melanoma where the tumor cell was engineered to express chicken Ova, and could be cleared by functional Ova-specific CD8 T cells. Using this system, mice vaccinated with Ova+AV were fully protected from tumor challenge, while Ova+PBS vaccinated mice were not (*Figure 1K*). Together, these data demonstrate that SE adjuvants can generate robust IgG responses and protective CD8 T cell responses.

### MF59 and AV elicit robust innate immune responses

In order to understand the mechanisms driving the robust adjuvant activity of SE adjuvants, we analyzed the innate immune responses in the draining lymph nodes at early time points following vaccination in both alum- and SE adjuvants-immunized groups (*Figure 2—figure supplement 1A*). SE groups displayed a significant increase of dLN size from 24 hr post-vaccination, while no increase in cellularity could be detected in alum-immunized dLNs even at 48 hr (*Figure 2A* and *Figure 2—figure supplement 1B*). The increase of dLN cellularity was due to the rapid recruitment of monocytes and neutrophils, followed by late influx of DCs and lymphocytes including B and T cells (*Figure 2B*). Next, we measured the capacity of antigen (Ag) uptake using fluorochrome-conjugated Ova (Ova-AF647) in various populations. Following vaccination, total LN cells in the SE groups showed higher Ag uptake when compared to LN cells from mock or alum groups (*Figure 2C* and *Figure 2—figure supplement 1C*). Neutrophils, monocytes and DCs were mainly responsible for Ag uptake at early time points, but DCs dominated later on (*Figure 2D*). Additionally, we examined the activation of DCs by measuring the surface expression of co-stimulatory molecules CD80 and CD86. At 24 hr post-vaccination, migratory DCs (mDCs) and resident DCs (rDCs) in both AV and MF59 groups had significantly increased the expression of CD80 and CD86 (*Figure 2E* and *Figure 2—figure supplement 1D*; *Idoyaga et al., 2013*). Kinetic analysis of CD80 expression on mDCs and rDCs showed a gradual increase in AV group over time, while no significant alteration was detected in alum group until 48 hr post-vaccination (*Figure 2F*). Taken together, SE adjuvants induce enhanced leukocyte recruitment, increased Ag uptake, and stronger DC activation, potentially explaining the potency of subsequent adaptive immune responses (*Calabro et al., 2011*).

### LN-resident macrophages are perturbed following MF59 and AV uptake

To examine how the SE adjuvants stimulate enhanced innate immune responses, we investigated the role of SE adjuvants in dLNs at early time points. Previous studies have established SIGN-R1$^-$$^+$ medullary macrophages within medullary inter-follicular regions as a primary collection point of

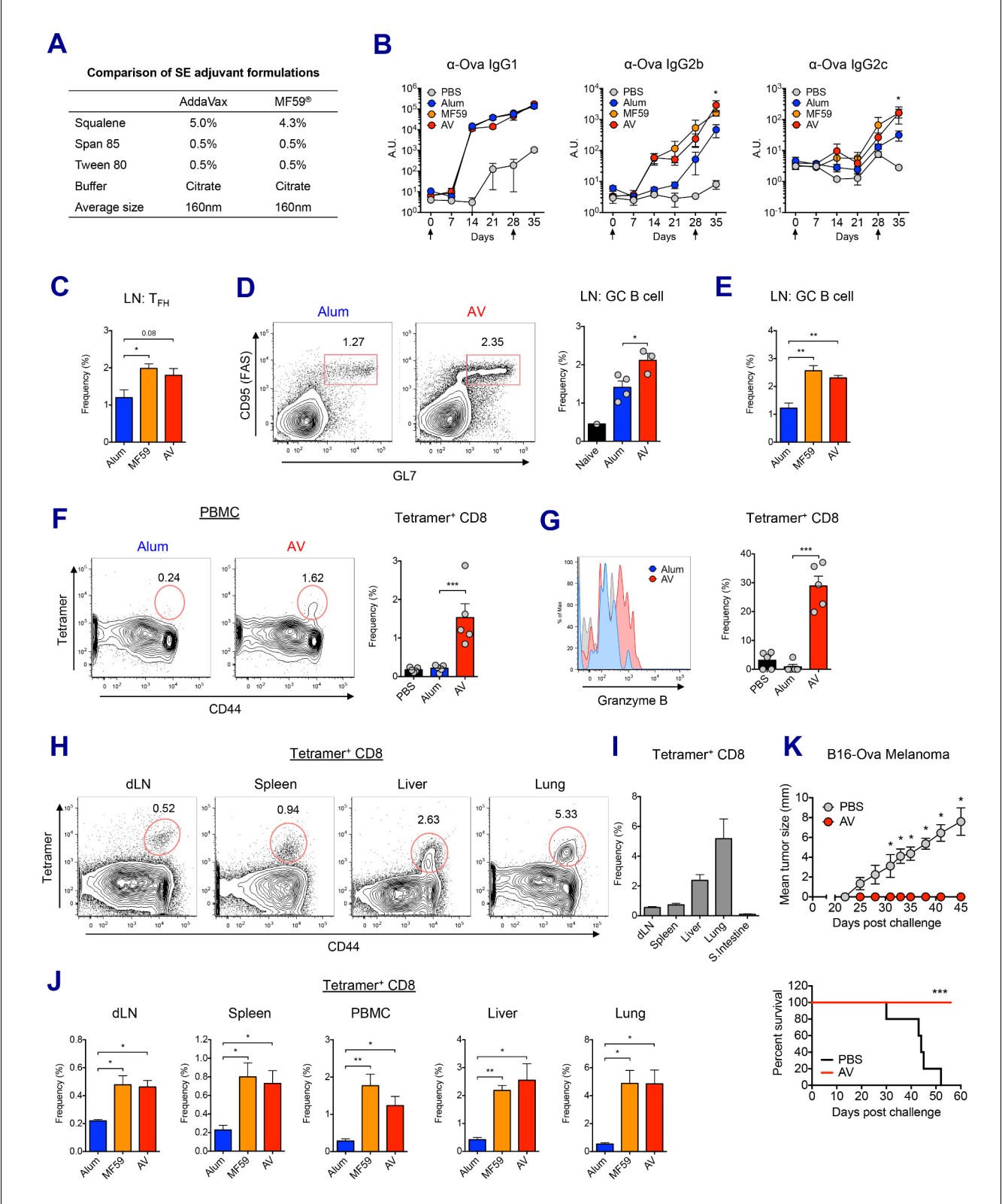

**Figure 1.** MF59 and AddaVax induce robust antibody and CD8 T cell responses. (**A**) Formulation comparison of two SE adjuvants; AddaVax and MF59 (**B**) WT B6 mice were primed and boosted with Ova mixed with MF59, AV or alum at indicated time points (arrows). Levels of Ova-specific IgG1, IgG2b and IgG2c in serum were determined by ELISA. *p<0.05 (*t*-test). (**C–J**) After the prime-boost vaccination with Ova plus MF59, AV or alum, immune responses were measured at day 7 post-boost. (**C**) Frequency of follicular helper T cells (CXCR5$^+$ PD1$^+$ CD4$^+$) *p<0.05 (*t*-test) (**D and E**) Frequency of

*Figure 1 continued on next page*

Figure 1 continued

germinal center B cells *p<0.05 (*t*-test). **p<0.01 (*t*-test) (**F**) Ova-specific CD8 T cell response was assessed by class I MHC-peptide tetramer staining. ***p<0.001 (*t*-test). (**G**) Granzyme B expression was measured among Ova-specific CD8 T cells. ***p<0.001 (*t*-test). (**H, I and J**) FACS plots and bar graphs displaying Ova-specific CD8 T cell responses in different organs. (**K**) 10 days after the prime-boost vaccination with Ova alone or Ova plus AV, WT B6 mice were challenged with Ova-expressing B16 melanoma. Graphs show size of tumor (upper panel) and survival of mice (lower panel). *p<0.05 (ANOVA). For the survival curve, p value was determined by Log-rank (Mantel-Cox) test. Data are representative of two to three independent experiments (mean and s.e.m.).

The online version of this article includes the following figure supplement(s) for figure 1:

**Figure supplement 1.** Experimental design and adoptive immune responses by alum, MF59 and AV.

lymph-borne antigen due to their proximal nature to the afferent lymphatics and high levels of phagocytosis (*Gray and Cyster, 2012*; *Woodruff et al., 2014*). Indeed, at 2 hr post-immunization, lipophilic dye-labeled AV could be detected primarily within the medullary region of the dLN

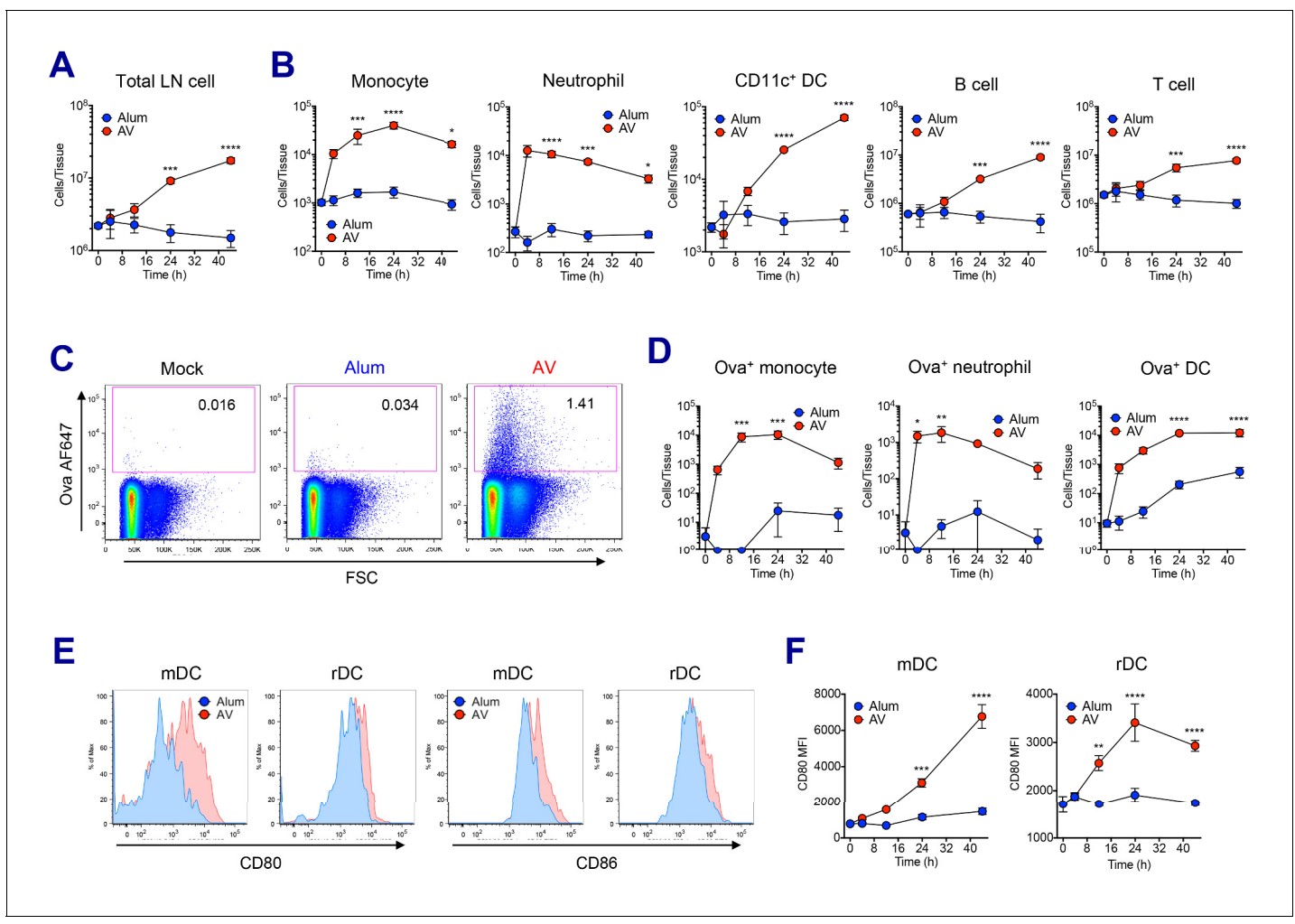

**Figure 2.** AV elicits robust innate immune responses. WT B6 mice were vaccinated with Ova mixed with AV or alum, and innate immune responses were assessed in dLNs during the first 48 hr period. (**A**) Total cell numbers of inguinal LNs. (**B**) Absolute numbers of different immune cell populations. (**C**) At 12 hr post-vaccination, the uptake of antigen was measured using AF647-conjugated Ova. (**D**) Total numbers of Ova$^+$ immune cell subsets were plotted. (**E**) Histograms show surface expression of CD80 and CD86 on migratory DCs and resident DCs. (**F**) Kinetic analysis of CD80 expression. Data are representative of two independent experiments (mean and s.e.m.). *p<0.05, **p<0.01, ***p<0.001, ****p<0.0001 (ANOVA).

The online version of this article includes the following figure supplement(s) for figure 2:

**Figure supplement 1.** Analysis of innate immune response.

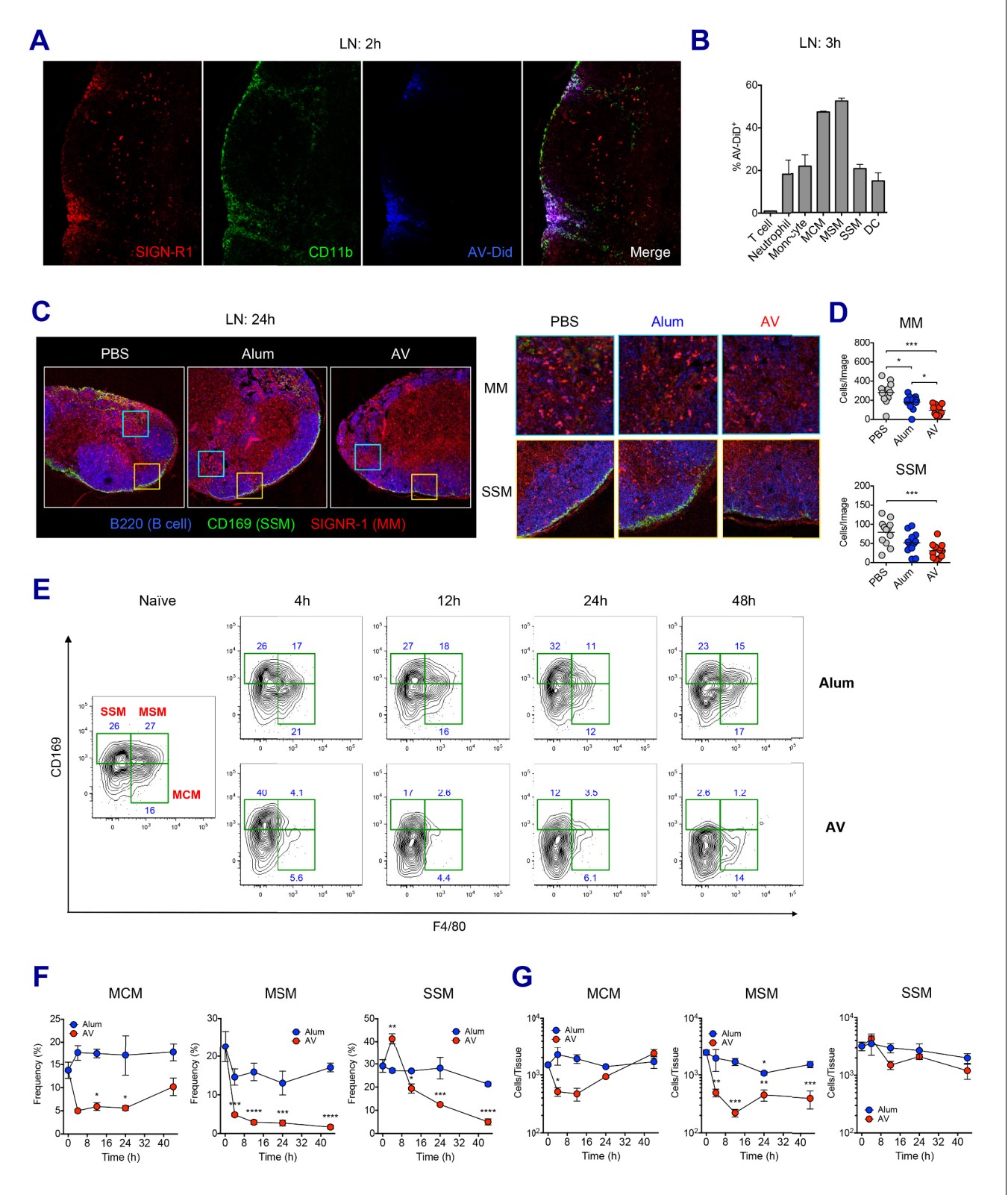

**Figure 3.** LN-resident macrophages uptake SE adjuvant and are eliminated. (**A and B**) WT B6 mice were vaccinated with Ova together with AV-Did, and the uptake of AV-Did was measured in dLNs by immunofluorescence at 2 hr (**A**) and flow cytometry at 3 hr (**B**). (**C and D**) Immunofluorescence of dLNs at 24 hr post-vaccination with PBS, alum or AV. (**D**) Quantification of MMs (upper) and SSMs (lower) from the acquired confocal images using CellProfiler software. *p<0.05, ***p<0.001 (ANOVA). (**E–G**) Kinetic changes of dLN-resident macrophage subsets were analyzed by flow cytometry.

*Figure 3 continued on next page*

*Figure 3 continued*

FACS plots are pre-gated on CD11b$^+$CD11c$^-$ Siglec-F$^-$ Ly6G$^-$ Ly6C$^-$ population. Frequencies (**F**) and absolute numbers (**G**) of each macrophage subset were plotted. *p<0.05, **p<0.01, ***p<0.001, ****p<0.0001 (ANOVA). Data are representative of two to three independent experiments (mean and s.e. m.).

The online version of this article includes the following figure supplement(s) for figure 3:

**Figure supplement 1.** LN-resident macrophages uptake MF59 and undergo elimination.

(*Figure 3A*). Flow cytometry analysis further verified that medullary cord macrophages (CD169$^-$ F4/80$^+$, MCMs) and medullary sinus macrophages (CD169$^+$ F4/80$^+$, MSMs) were the most abundantly represented cell types (up to 50%) within the AV$^+$ populations, although lower frequencies of other innate immune cells including neutrophils, monocytes, DCs and subcapsular sinus macrophages (CD169$^+$ F4/80$^-$, SSMs) were also AV$^+$ (*Figure 3B*).

Next, we wondered if the homeostasis of macrophages in dLN were affected by the uptake of SE adjuvants. To this end, the presence of medullary macrophages (MMs) and SSMs in differently immunized dLNs were assessed by confocal microscopy. At 24 hr after vaccination, PBS-injected dLNs displayed abundant MMs and a clear lining of SSMs along the subcapsular sinus. However, dLNs in the AV and MF59 groups displayed a remarkable decrease of both MM and SSM populations with an almost complete loss of the subcapsular sinus lining, while there were no such reductions in mice immunized with alum plus Ova (*Figure 3C and D* and *Figure 3—figure supplement 1A*). To dissect this phenomenon further, we performed a time course analysis on macrophage populations in dLNs from alum and AV groups during the first 2 days post-vaccination by flow cytometry. Consistent with confocal data, alum did not cause significant alteration in any of the macrophage subsets, even until 48 hr post-immunization (*Figure 3E and F*). However, in the AV group at 4 hr and 12 hr post-vaccination, there was a striking reduction in the frequencies of both types of medullary macrophages (MCMs and MSMs). Moreover, there was a reduction in the frequencies of SSM albeit with slower kinetics, starting at 12 hr and dropping to the lowest levels at 48 hr (*Figure 3E and F*). Similarly, MF59 also caused a significant reduction of LN-resident macrophages at 24 hr post-immunization (*Figure 3—figure supplement 1B and C*). In terms of absolute number, MSMs displayed a significant and persistent reduction, while MCMs were reduced transiently and SSMs displayed a mild reduction (*Figure 3G*). The discrepancy between frequencies and absolute numbers is explained by the dramatic increase of dLN cellularity after 24 hr post-immunization (*Figure 2A*). Collectively, these data show that the efficient phagocytosis of SE adjuvants is associated with the early loss of MMs including MCMs and MSMs, while the relatively slow accumulation of AV in SSMs is associated with the delayed and less pronounced loss.

## MF59 and AV trigger sequential waves of regulated necrosis and apoptosis in LN macrophages

We sought to understand whether the loss of LN-resident macrophages was due to cell death or any other mechanism. Using bone marrow-derived macrophages (BMMs), the kinetics of in vitro cell death was assessed in the presence of AV. AV induced dynamic changes such as dominant necrosis (AnnexinV$^-$ PI$^+$) at early time points (up to 6 hr), followed by delayed apoptosis (AnnexinV$^+$ PI$^-$) (*Figure 4A*). In order to examine if this is the case in vivo, dLNs were collected after vaccination with alum or AV, and subjected to western blot of whole LN lysate. Consistent with the in vitro cell death data, phosphorylated-MLKL (p-MLKL) and cleaved caspase 1 (markers of necroptosis and inflammasome activation/pyroptosis respectively) were visible at 3 hr, and were strongly induced at 6 hr in AV-immunized dLNs (*Figure 4B*). Immuno-fluorescent staining of phosphorylated RIPK3 and MLKL verified necroptosis signaling in the medullar following AV immunization (*Figure 4C*). Induction of MLKL phosphorylation was confirmed to be RIPK3-dependent as no positive p-MLKL signal could be observed within AV-immunized RIPK3-KO animals (*Figure 4—figure supplement 1A*). Meanwhile no MLKL phosphorylation and lower levels of cleaved caspase 1 were detected in alum-immunized dLNs. Furthermore, there was increased apoptosis in the dLNs of mice immunized with alum, as evidenced by enhanced cleavage of caspase 3 at 3 hr and 6 hr, while AV injection displayed a strong activation of caspase 3 at 24 hr (*Figure 4B*). These cell death phenotypes in dLNs were also true for MF59 vaccination (*Figure 4—figure supplement 1B*). Thus, although these data do not give

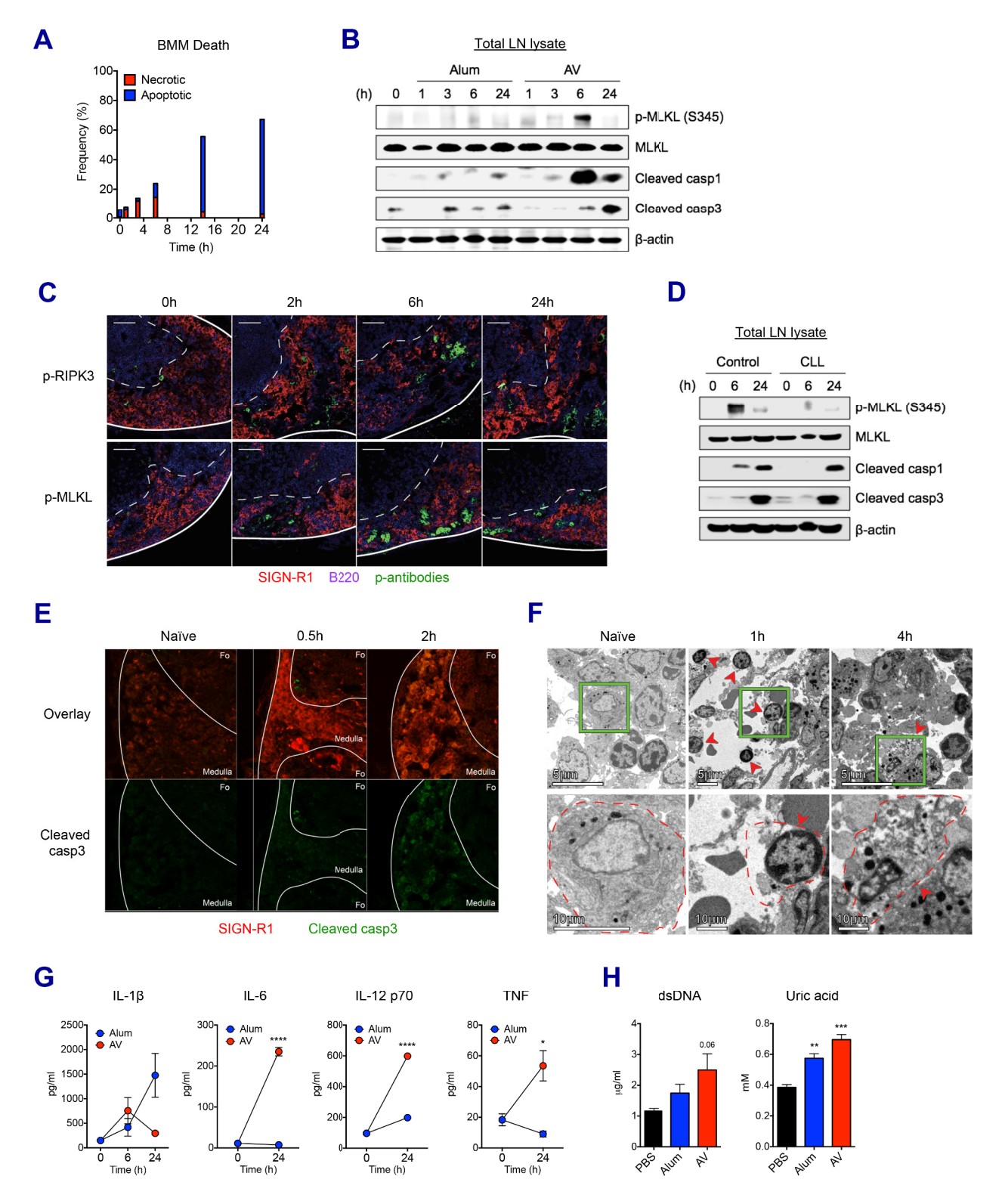

**Figure 4.** SE adjuvant triggers sequential waves of regulated necrosis and apoptosis in LN macrophages. (**A**) Bone marrow-derived macrophages were incubated with AV for different time periods. Using co-staining of AnnexinV and Propidium Iodide, kinetic changes of necrotic and apoptotic cells were determined by flow cytometry. (**B**) After WT B6 mice were immunized with alum or AV, dLNs from each group were collected, and total dLN lysates were subjected to western blot analysis at indicated time points. (**C**) Immunofluorescence of dLNs at indicated time points to detect necroptosis

*Figure 4 continued on next page*

*Figure 4 continued*

signaling. (D) Western blot analysis with dLN lysates from control and CLL-treated WT B6 mice. (E) Immunofluorescence of dLNs at 0 hr, 0.5 hr and 2 hr post-vaccination. (F) Electron micrographs displaying AV-vaccinated dLNs at different time points. Red arrows represent cells with necrotic phenotype. (G) Serum levels of different cytokines were assessed by ELISA. *p<0.05, ****p<0.0001 (ANOVA). (H) At 24 hr post-immunization, the release of dsDNA and uric acid was determined in serum. **p<0.01, ***p<0.001 (*t*-test). Data are representative of two independent experiments (mean and s.e.m.). The online version of this article includes the following figure supplement(s) for figure 4:

**Figure supplement 1.** SE adjuvants triggers cell death signaling pathways.

---

information on which cell types are undergoing cell death, they suggest that after SE-adjuvanted immunization, there is early regulated necrosis such as MLKL-mediated necroptosis and pyroptosis, followed by delayed apoptosis in dLNs.

Clodronate-loaded liposomes (CLL) are known to induce selective apoptosis in macrophages (*van Rooijen and Hendrikx, 2010*). Consistent with the literature, subcutaneous injection of CLL resulted in efficient depletion of MSM and SSM subsets in dLNs (*Figure 4—figure supplement 1C*). Interestingly, following macrophage depletion, Ova+AV vaccination resulted in significantly decreased MLKL phosphorylation and altered kinetics of caspase1 cleavage (*Figure 4D*). These data, combined with the observed loss of the LN-resident macrophage populations by both flow cytometry and immunofluorescence (*Figure 3C–G*) suggests that LN-resident macrophages contribute substantially to the observed necroptosis in dLNs after AV vaccination. In addition to putative LN-resident macrophages undergoing necroptosis, MMs displaying cleaved caspase 3 were also detected after the AV vaccination (*Figure 4E*). Morphologically, necrotic cell death is characterized by organelle swelling, membrane disruption, and cytoplasmic release as opposed to the chromatin condensation and membrane blebbing phenotype typical of apoptosis. In contrast to the minimal level of dying cells in dLNs of naïve mice, necrotic cells (red arrows), including necrotic monocytes/macrophages (1 hr) and a necrotic macrophage (4 hr) were observed predominantly at the medulla of dLNs after 1 hr of AV injection (*Figure 4F*). Together these data consistently support the idea that LN-resident macrophages undergo early necrosis and delayed apoptosis upon AV immunization.

An extensive literature has outlined the role of necrosis in the generation of inflammatory responses through the secretion of pro-inflammatory cytokines and the released danger signals that can act as DAMPs (*Bergsbaken et al., 2009*; *Blander, 2014*; *Mocarski et al., 2014*; *Pasparakis and Vandenabeele, 2015*). Therefore, we asked if these signals could be detected systemically following AV vaccination. Indeed, cytokines such as IL-1β, IL-6, IL-12 and TNF were released into serum by AV vaccination (*Figure 4G*). IL-6, IL-12 and TNF were particularly elevated by AV vaccination in comparison with the alum group. Moreover, elevated levels of double-stranded DNA (dsDNA) and uric acid, well described DAMPs could be increasingly detected in serum of AV-immunized mice at 24 hr (*Figure 4H*; *Ko et al., 2016*). Collectively, these data indicate that the subcutaneous injection of SE adjuvants can induce mixed types of cell death signaling pathways which in turn set up a highly immuno-stimulatory microenvironment.

## Macrophages and Batf3$^+$ DCs cooperatively stimulate CD8 T cell response upon SE-adjuvanted immunization

To determine the importance of macrophage perturbation in the induction of innate and adaptive immune responses, both pharmacological and genetic approaches were used to deplete LN-resident macrophages. Following macrophage depletion by CLL treatment, Ova+AV vaccination resulted in reduced activation of the mDC subset, compared to control liposome treatment at 24 hr (*Figure 5—figure supplement 1A*). Production of IL-6 was also dramatically attenuated (*Figure 5—figure supplement 1B*). Importantly, Ova-specific CD8 T cell responses were significantly decreased in the CLL-treated group at day 7 post-boost (*Figure 5A*).

In order to more specifically dissect the effect of macrophage population on CD8 T cell responses, we used two different genetic models of macrophage depletion: CD169-DTR and LysM-iDTR mice. Upon intraperitoneal DT injection, CD169-DTR mice showed efficient depletion in MSM and SSM subsets, while LysM-iDTR mice displayed only SSM depletion (*Figure 5B* and *Figure 5—figure supplement 1C*; *Gupta et al., 2016*; *Shaabani et al., 2016*). Surprisingly, in contrast to

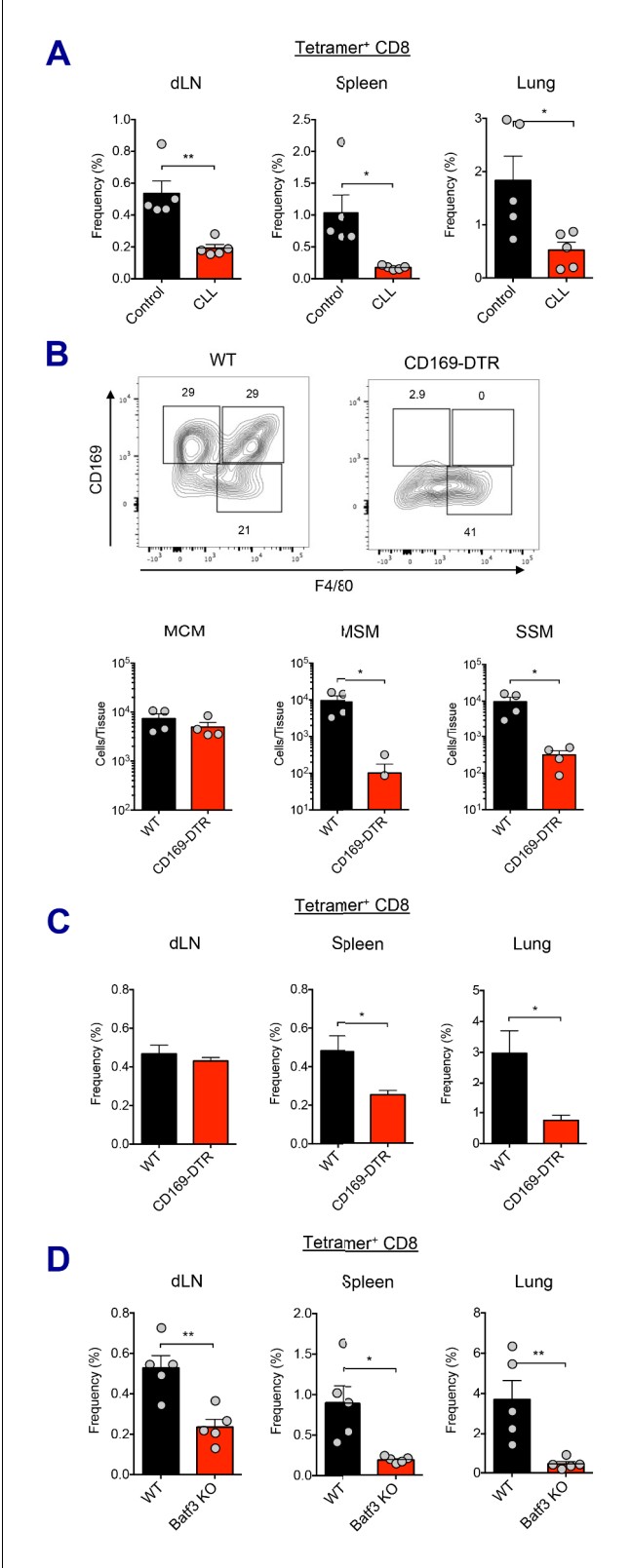

**Figure 5.** Macrophages and Batf3[+] DCs cooperatively stimulate CD8 T cell response. (**A**) WT B6 mice were injected with control or clodronate-loaded liposomes. 5 days later, mice were subsequently immunized with Ova and AV. Ova-specific CD8 T cell response at day 7 post-boost in different tissues. *p<0.05, **p<0.01 (*t*-test). (**B and C**) Mice got intraperitoneal injection with 400 ng of diphtheria toxin per mouse, and two days later, they were

*Figure 5 continued on next page*

*Figure 5 continued*

subsequently immunized by Ova plus AV. (**B**) The depletion of LN-resident macrophages was measured in CD169-DTR mice two days after the DT injection. *p<0.05 (*t*-test). (**C**) Ova-specific CD8 T cell responses in WT B6 and CD169-DTR mice at day 7 post-boost. *p<0.05 (*t*-test). (**D**) Ova-specific CD8 T cell response in WT B6 and Batf3 KO mice. *p<0.05, **p<0.01 (*t*-test). Data are representative of two independent experiments (mean and s.e.m.). The online version of this article includes the following figure supplement(s) for figure 5:

**Figure supplement 1.** Association of macrophages and dendritic cells in SE adjuvant-mediated immune responses.

---

normal CD8 T cell responses in the LysM-iDTR group (*Figure 5—figure supplement 1D*), the CD169-DTR group showed significant reduction of CD8 T cell responses in spleen and lung (*Figure 5C*). These data demonstrate that LN macrophages are required for the efficient activation of innate immune response and the optimal induction of CD8 T cell response by SE adjuvants. Of note, the recent description of a conventional DC2 (cDC2, CD11b$^+$ CD11c$^+$) population opens the possibility of an additional actor in this response pathway (*Ciavarra et al., 2005*). While lymph node macrophages have been directly tested here, it is unclear if the cDC2 population might also play a role as their function has not yet been fully validated across these various model systems.

With respect to the cell types involved in Ag cross-presentation to CD8 T cells, it is known that CD8a$^+$ DCs are the key mediators of this process (*den Haan et al., 2000*; *Hildner et al., 2008*). It has been reported that Batf3$^+$ DCs such as Xcr1$^+$ CD103$^+$ DCs essential for cross-presentation of skin-derived Ags to CD8 T cells (*Bedoui et al., 2009*). In contrast, a recent study suggested that CD169$^+$ macrophages are capable of Ag cross-presentation to CD8 T cells without DCs in specific experimental settings (*Bernhard et al., 2015*). To investigate which cell type is responsible for Ag cross-presentation to CD8 T cells in the context of the current study, we asked if Batf3$^+$ DCs were required for the AV-mediated CD8 T cell responses using Batf3-deficient mice. In the absence of Batf3$^+$ DC populations (*Figure 5—figure supplement 1E*), the Ova-specific CD8 T cell response was substantially impaired (*Figure 5D*). This implies that cross-presentation by Batf3$^+$ DCs is critical for the generation of CD8 T cell responses in SE-adjuvanted vaccination. Taken together, these data demonstrate that LN macrophages and Batf3-dependent CD103$^+$ DCs are necessary for optimal induction of antigen-specific CD8 T cell response to SE adjuvants.

## Kinase activity of RIPK3 in macrophages is critical for the induction of CD8 T cell response by SE adjuvants

We sought to understand the molecular pathways governing the CD8 T cell response in SE-adjuvanted vaccination model. To this end, TLR signaling pathways were investigated utilizing different KO mice. Surprisingly, neither individual TLR deficiency (TLR3, TLR4, TLR7, and TLR9) nor the absence of adaptor proteins such as MyD88 and TRIF affected AV-induced CD8 T cell response. (*Figure 6—figure supplement 1A and B*). Since SE adjuvants induce RIPK3-dependent macrophage death (*Figure 4D* and *Figure 4—figure supplement 1A*), and dLN-resident macrophages are necessary for CD8 T cell responses (*Figure 5A-C*), we investigated whether different types of cell death pathways are involved in this process. To probe the role of inflammasomes and pyroptosis, we examined the CD8 T cell response in ASC KO and Caspase 1 KO mice and observed that the CD8 T cell response was normal in these mice (*Figure 6—figure supplement 1C*). In addition, in vivo inhibition of pan-caspase activity by z-VAD-fmk did not influence the normal CD8 T cell response (*Figure 6—figure supplement 1D*), suggesting that neither inflammasome activation/pyroptosis nor apoptosis is crucial for AV-induced CD8 T cell responses. A previous report by Yatim et al. demonstrated that an injection of necroptotic cells rather than apoptotic cells generated potent CD8 T cell response (*Yatim et al., 2015*). To elucidate the role of RIPK3 signaling in SE-adjuvanted vaccination, we assessed immune responses in RIPK3 KO mice upon Ova+AV vaccination. At 24 hr after vaccination, RIPK3 KO mice displayed significantly impaired secretion of IL-6 and TNF in serum (*Figure 6A*). Remarkably, in the absence of RIPK3, CD8 T cell response in various tissues was significantly reduced by AV vaccination, as compared to WT mice (*Figure 6B*). Consistent with AV data, vaccination with MF59 in RIPK3 KO mice resulted in a significant decrease of CD8 T cells (*Figure 6—figure supplement 2A*).

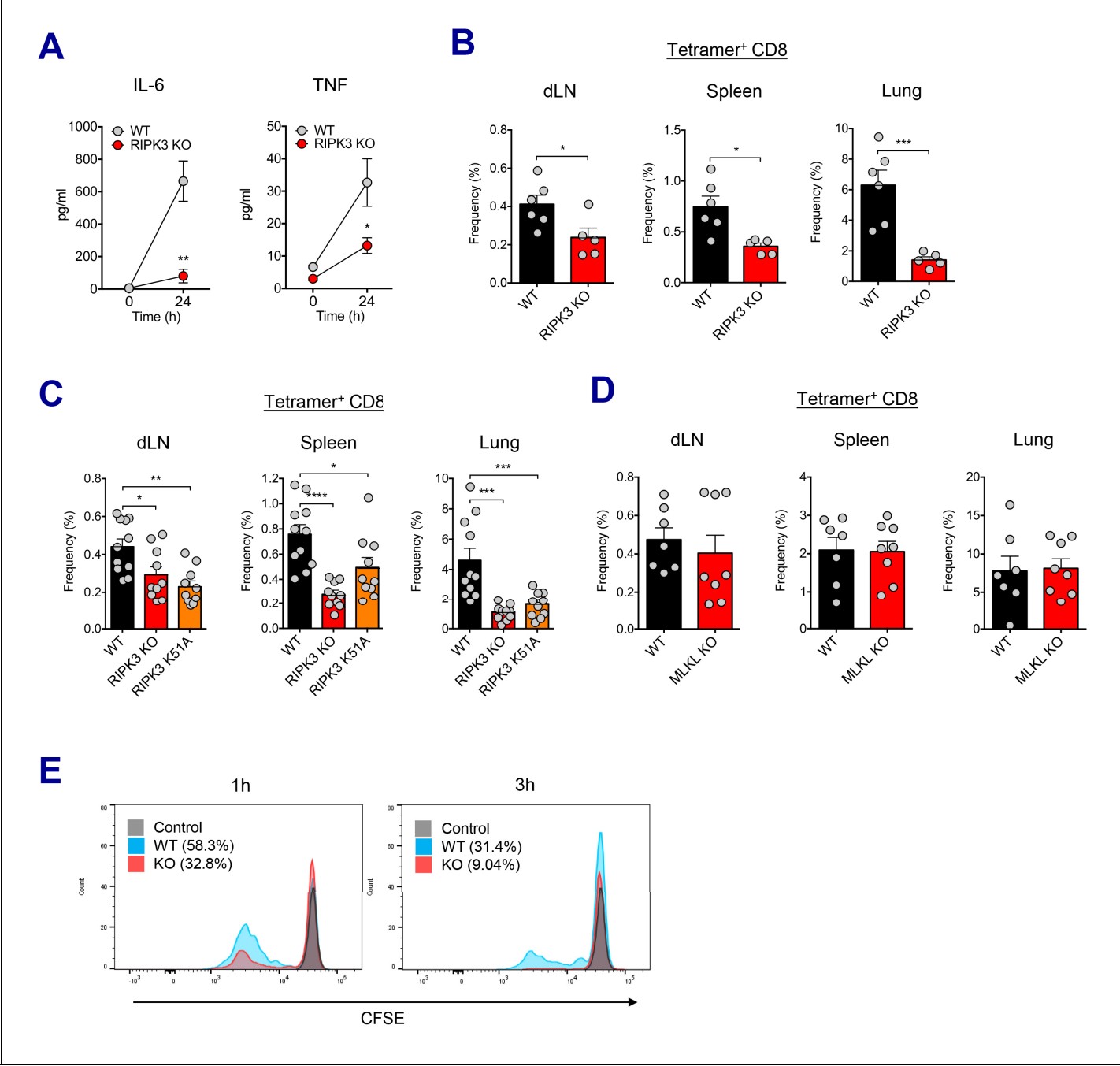

**Figure 6.** RIPK3 kinase-dependent signaling is critical for the induction of CD8 T cell response. (**A**) WT B6 and RIPK3 KO mice were primed with Ova and AV, and serum cytokine levels were quantified at 24 hr. (**B**) At day 7 post-boost, Ova-specific CD8 T cell response was determined from different tissues in WT B6 and RIPK3 KO mice. *p<0.05, ***p<0.001 (*t*-test). (**C**) Ova-specific CD8 T cell responses in WT B6, RIPK3 KO and RIPK3 K51A kinase-dead mice. Data were pooled from two independent experiments. *p<0.05, **p<0.01, ***p<0.001, ****p<0.0001 (*t*-test). (**D**) Ova-specific CD8 T cell responses in WT B6 and MLKL KO mice. (**E**) Histograms represent OT-I cell proliferation upon Ag cross-presentation (1h- or 3h-incubation of BMMs with AV). Gray, blue and red colors represent control, WT B6 BMM and RIPK3 KO BMM groups, respectively. Data are representative of two to three independent experiments (mean and s.e.m.).

The online version of this article includes the following figure supplement(s) for figure 6:

**Figure supplement 1.** AV-induced CD8 T cell responses in different KO mice and drug treatment.

**Figure supplement 2.** CD8 T cell responses in the absence of RIPK3 by SE adjuvants.

Recent studies have proposed multiple roles for the RIPK3 protein – the kinase-dependent functions including the execution of necroptosis and induction of inflammatory cytokines, and kinase-independent scaffolding functions (*Kang et al., 2013*; *Lawlor et al., 2015*; *Muendlein et al., 2020*; *Najjar et al., 2016*; *Zhu et al., 2018*). To further dissect the role of kinase activity of RIPK3 in the SE-adjuvanted immunization, we utilized a RIPK3 kinase-inactive knock-in mutant (*Ripk3*$^{K51A/K51A}$) mouse model. RIPK3 K51A mice were also significantly impaired in their capacity to induce Ova-specific CD8 T cell responses in multiple tissues, albeit to a lesser degree than RIPK3 KO mice (*Figure 6C*), suggesting that the kinase-dependent activity of RIPK3 is necessary for induction of CD8 T cell response by SE adjuvants. Next, we examined whether the execution of necroptosis is essential for the AV-induced CD8 T cell response. Notably, deficiency of MLKL, an effector molecule of necroptosis, did not affect Ova-specific CD8 T cell response after Ova+AV vaccination (*Figure 6D*), implying an alternate pathway for RIPK3 function in this system outside of the classically described execution of necroptosis.

To directly address if RIPK3-dependent signaling in macrophages is necessary for the induction of CD8 T cell response, we designed an in vitro cross-presentation assay in which bone marrow-derived macrophages (BMMs) incubated with Ova and AV were subsequently co-cultured with DCs, followed by the addition of CFSE-labeled OT-I cells (*Figure 6—figure supplement 2B*). Strikingly, BMMs from the RIPK3 KO mouse resulted in the significantly decreased proliferation of OT-I cells, as compared to WT BMMs (*Figure 6E*). This result was observed regardless of different duration (1 hr and 3 hr) of SE adjuvant incubation with BMMs. Therefore, these data demonstrate that RIPK3-dependent signaling in dLN-resident macrophages is necessary for optimal CD8 T cell responses to SE-adjuvanted immunization.

## IgG responses by SE adjuvants are not dependent on the RIPK3 signaling pathway

Vaccine-induced antibody responses are essential for protective immunity. In addition to the previously described CD8 T cell response here, SE adjuvants are known to elicit strong humoral immune responses, consistent with the results observed in the current study (*Figure 1B*). In order to understand how these adjuvants trigger potent antibody responses, we determined if dLN-resident macrophages are also associated with AV-induced IgG responses. Consistent with the role of macrophages in CD8 T cell responses, the depletion of both SSM and MSM in CD169-DTR mice resulted in the significant reduction of Ova-specific IgG responses (*Figure 7A*). In contrast, there were normal IgG responses in CLL-treated mice and LysM-iDTR mice (Data not shown and *Figure 7—figure supplement 1A*, respectively). This phenotypic discrepancy between CD8 T cell response and IgG response in CLL-treated mice could be due to CLL's direct adjuvant effect on B cells (*Tonti et al., 2013*). Furthermore, the absence of Batf3$^+$ DCs did not alter IgG responses (*Figure 7—figure supplement 1B*).

Next, we explored the effect of different types of cell death on IgG responses. In contrast to the CD8 T cell response, in vivo treatment with the pan-caspase inhibitor, z-VAD-fmk, resulted in a profound decrease in antigen-specific IgG1 titers induced by AV (*Figure 7B*). However, the inhibition of RIPK1 by Nec-1s injection or RIPK3 deficiency had no effect on IgG responses in AV vaccination (*Figure 7B and C*). In addition, impaired inflammasome formation in ASC KO and caspase 1 KO mice resulted in normal IgG responses (*Figure 7—figure supplement 1C*). This suggests that caspase-dependent apoptosis, rather than a RIPK3-dependent pathway might be primarily important for antibody responses induced by AV vaccination. Intriguingly, previous studies have reported that the modulation of DAMPs such as dsDNA in various vaccination models affects humoral immune response (*Marichal et al., 2011*). In the AV-adjuvanted vaccination model, the efficient generation of IgG responses was also associated with the release of dsDNA since the co-injection of DNaseI significantly decreased the IgG responses by AV (*Figure 7D*). In alignments with a previous report (*Seubert et al., 2011*), the MyD88 pathway was required for the generation of optimal IgG responses by SE adjuvant although some of individual TLRs are moderately associated (*Figure 7—figure supplement 1D and E*). Altogether, these data suggest that the caspase activity in macrophages and subsequent release of DAMPs might play important roles in the SE adjuvant-mediated IgG responses.

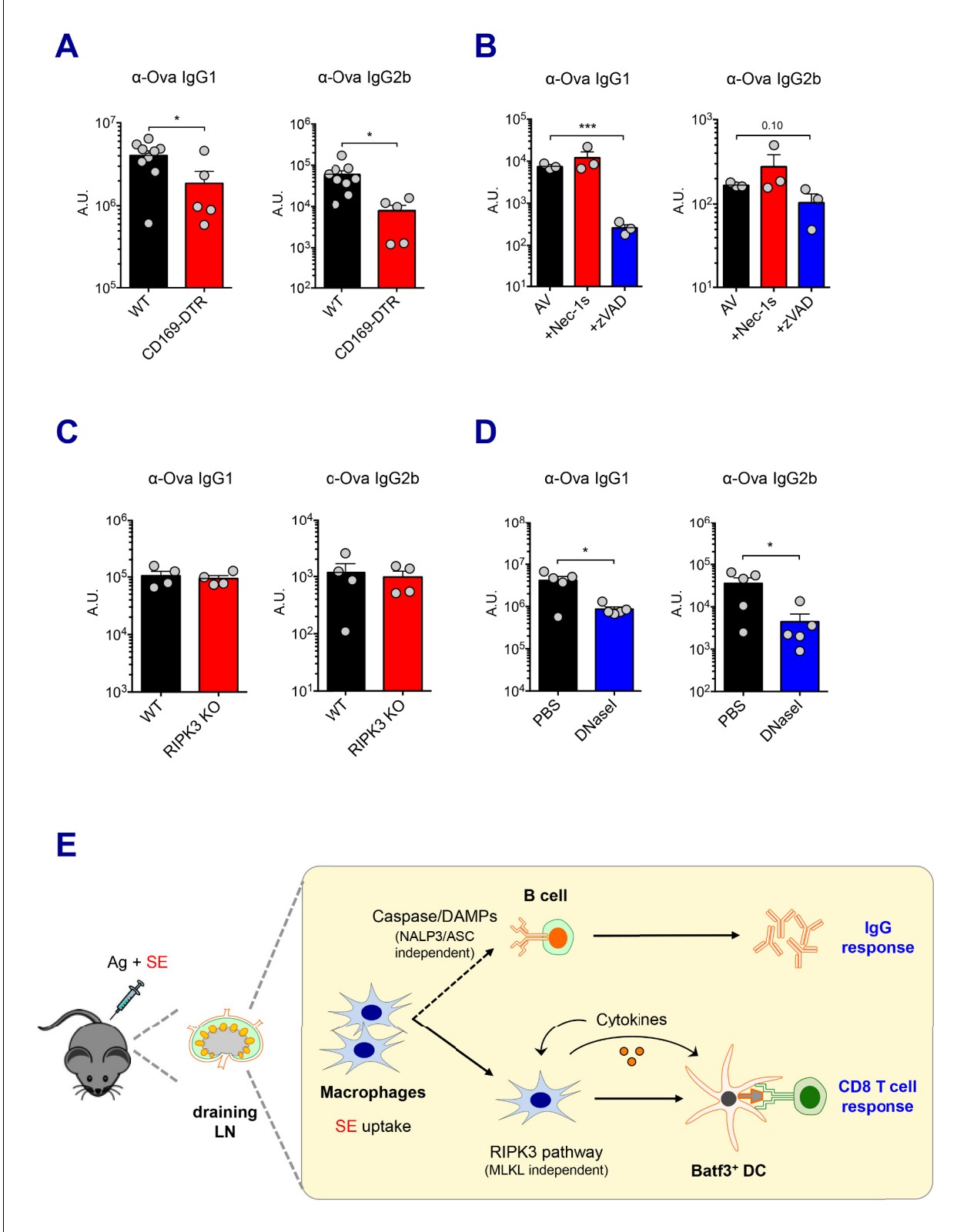

**Figure 7.** Caspase activity and DAMPs promote IgG responses. Levels of IgG1 and IgG2b were assessed in serum samples at day 7 post-boost. (**A**) IgG levels in WT B6 and CD169-DTR mice at day 7 post-boost. (**B**) WT B6 mice were immunized with AV in the presence or absence of Nec-1s or z-VAD-fmk. (**C**) IgG levels in WT B6 and RIPK3 KO mice. (**D**) WT B6 mice were primed and boosted with AV in the presence or absence of DNaseI. (**E**) A
*Figure 7 continued on next page*

*Figure 7 continued*

working model describing distinctive mechanisms governing the IgG and CD8 T cell responses induced by the SE adjuvant. *p<0.05, ***p<0.001 (*t*-test). Data are representative of two to three independent experiments (mean and s.e.m.).

The online version of this article includes the following figure supplement(s) for figure 7:

**Figure supplement 1.** Ova-specific antibody responses.

## Discussion

Recently cellular stress and damage have become increasingly recognized as potent drivers of inflammation and the adaptive immune response (*Bettigole and Glimcher, 2015*; *Blander, 2014*; *Chovatiya and Medzhitov, 2014*; *Matzinger, 1994*; *Osorio et al., 2014*; *Pulendran, 2015*; *Ravindran et al., 2014*; *Zhang and Kaufman, 2008*). It has also become clear that cell death and associated signaling pathway can modulate immune outcomes (*Bergsbaken et al., 2009*; *Blander, 2014*; *Chan et al., 2015*; *Murakami et al., 2014*; *Pasparakis and Vandenabeele, 2015*). For example, apoptosis in the absence of foreign antigen (Ag) is generally considered as immunologically silent, while regulated necrosis such as RIPK3-dependent necroptosis and pyroptosis results in the release of pro-inflammatory cytokines and DAMPs (*Kim et al., 2019*; *Kolb et al., 2017*; *Berghe et al., 2014*). Injection of necroptotic cells rather than apoptotic cells promoted potent CD8 T cell responses in the presence of foreign Ag although it is noteworthy that death-associated signaling rather than necroptosis itself was critical (*Yatim et al., 2015*). This 'sterile inflammation' concept has revitalized the autoimmunity and oncology fields, but little is yet known about the role of these pathways in regulating the immune system in the context of vaccine response.

In this study, we present evidence that induction of antigen-specific CD8 T cell responses in mice immunized with MF59 or its mimetic AV occurs through a mechanism dependent on RIPK3 signaling in LN macrophages. Using several independent models, we demonstrate that the SE adjuvants are primarily acquired by LN-resident macrophages, resulting in their elimination through various cell death pathways including apoptosis and regulated necrosis. These infection/vaccination-mediated damage response doesn't seem to be rare event because recent studies have demonstrated that *Staphylococcus aureus* infection, Modified Vaccinia Ankara (MVA) vaccination or the saponin based adjuvant caused the disruption of SSM layer in dLNs, which in turn significantly modulate subsequent B cell or T cell responses, respectively (*Detienne et al., 2016*; *Gaya et al., 2015*; *Sagoo et al., 2016*). In this SE adjuvant vaccination model, it is worth noting that the MSM subset, rather than SSMs, is critical for the uptake of the SE adjuvant, inducing cell death-associated signaling, and ultimately required for optimal adaptive immune responses. Importantly, we demonstrate the induction of optimal antigen-specific antibody responses by SE adjuvants is associated with caspase activity and dsDNA release (*Figure 7E*). Thus, while RIPK3-dependent signaling is critical for optimal CD8 T cell response, it was not necessary for IgG responses.

RIPK3 is known to be essential in the induction of MLKL-dependent necroptosis, but recent studies have established additional roles for RIPK3 in kinase-dependent cytokine induction and kinase-independent scaffolding function (*Kang et al., 2013*; *Lawlor et al., 2015*; *Muendlein et al., 2020*; *Najjar et al., 2016*; *Zhu et al., 2018*). We demonstrate that the kinase activity of RIPK3 was critical for SE adjuvant-mediated CD8 T cell response since RIPK3 kinase-dead (RIPK3 K51A) mice abrogated the response. Further investigation using MLKL-deficient mice has revealed the dispensability of the execution step of necroptosis in eliciting CD8 T cell response although RIPK3's kinase activity is still required. Thus we propose that the increased secretion of proinflammatory cytokines by RIPK3-associated signaling is essential for the optimal CD8 T cell response to SE-adjuvanted immunization (*Snyder et al., 2019*; *Yatim et al., 2015*). In contrast, our data suggest that caspase activity is necessary for the efficient induction of IgG response by SE adjuvants. Caspase family proteins have two well-known functions, as triggers of apoptosis and in the activation of inflammasomes, depending on the caspase family member. The redundant role of the canonical inflammasome pathway in SE adjuvant-mediated IgG responses was demonstrated using caspase 1 and ASC deficient mouse models, raising the possibility of additional caspases' roles in the AV-induced antibody response such as apoptosis induction or previously unappreciated function.

Our data also demonstrate that the SE-adjuvanted immunization establishes a highly immuno-stimulatory environment. We observed increased secretion of pro-inflammatory cytokines and

DAMPs such as IL-6, IL-12 and TNF, and dsDNA by the SE adjuvant. It is possible that the release of pro-inflammatory mediators provides a feed-forward mechanism in the further elimination of macrophages and the efficient activation of DCs.

In conclusion, our results demonstrate that the CD8 T cell response, by MF59 and its analog AV, can be triggered via RIPK3-dependent signaling in dLN macrophages, whilst antibody responses occur independently of this pathway. These results suggest that pharmacological or genetic manipulation of these pathways may provide novel mediators of vaccine immunity.

# Materials and methods

## Key resources table

| Reagent type (species) or resource | Designation | Source or reference | Identifiers | Additional information |
|---|---|---|---|---|
| Genetic reagent (*M. musculus*) | C57BL/6J | The Jackson Laboratory | Stock No: 000664 | |
| Genetic reagent (*M. musculus*) | Batf3 KO | The Jackson Laboratory | Stock No: 013755 | |
| Genetic reagent (*M. musculus*) | Caspase 1 KO | The Jackson Laboratory | Stock No: 016621 | |
| Genetic reagent (*M. musculus*) | TLR3 KO | The Jackson Laboratory | Stock No: 005217 | |
| Genetic reagent (*M. musculus*) | TRIF KO | The Jackson Laboratory | Stock No: 005037 | |
| Genetic reagent (*M. musculus*) | RIPK3 KO | PMID:14749364 | | |
| Genetic reagent (*M. musculus*) | RIPK3$^{K51A/K51A}$ | PMID:25459880 | | |
| Genetic reagent (*M. musculus*) | CD169-DTR | PMID:23601688 | | |
| Genetic reagent (*M. musculus*) | LysM-iDTR | PMID:20176743 | | |
| Genetic reagent (*M. musculus*) | ASC KO | PMID:15190255 | | |
| Genetic reagent (*M. musculus*) | MLKL KO | PMID:24012422 | | |
| Genetic reagent (*M. musculus*) | TLR4 KO | PMID:16461338 | | |
| Genetic reagent (*M. musculus*) | TLR7 KO | PMID:16461338 | | |
| Genetic reagent (*M. musculus*) | TLR9 KO | PMID:20962088 | | |
| Genetic reagent (*M. musculus*) | MyD88 KO | PMID:16461338 | | |
| Antibody | CD11c (Hamster monoclonal) | BioLegend | 117330 | FCM 1:400 |
| Antibody | CD169 (Rat monoclonal) | BioLegend | 142413 | FCM 1:400 |
| Antibody | CD19 (Rat monoclonal) | BioLegend | 115541 | FCM 1:800 |
| Antibody | TCR-b (Hamster monoclonal) | BioLegend | 109224 | FCM 1:400 |

*Continued on next page*

*Continued*

| Reagent type (species) or resource | Designation | Source or reference | Identifiers | Additional information |
|---|---|---|---|---|
| Antibody | CD4 (Rat monoclonal) | BioLegend | 100557 | FCM 1:800 |
| Antibody | CD44 (Rat monoclonal) | BioLegend | 103047 | FCM 1:800 |
| Antibody | CD45R (B220) (Rat monoclonal) | BioLegend | 103244 103229 | FCM 1:800 IF 1:400 |
| Antibody | CD80 (Hamster monoclonal) | BioLegend | 104729 | FCM 1:800 |
| Antibody | CD86 (Rat monoclonal) | BioLegend | 105006 | FCM 1:800 |
| Antibody | CD8a (Rat monoclonal) | BioLegend | 100750 | FCM 1:800 |
| Antibody | IFN-g (Rat monoclonal) | BioLegend | 505806 | FCM 1:800 |
| Antibody | Ly6G (Rat monoclonal) | BioLegend | 127624 | FCM 1:800 |
| Antibody | Rabbit IgG AF647 (Donkey polyclonal) | BioLegend | 406421 | IF 1:400 |
| Antibody | Fc block (Rat monoclonal) | BD Biosciences | 553142 | FCM 1:800 |
| Antibody | CD103 (Rat monoclonal) | BD Biosciences | 564322 | FCM 1:400 |
| Antibody | CD11b (Rat monoclonal) | BD Biosciences | 563402 | FCM 1:800 |
| Antibody | CD279 (PD-1) (Hamster monoclonal) | BD Biosciences | 563059 | FCM 1:200 |
| Antibody | CD45 (Rat monoclonal) | BD Biosciences | 563053 | FCM 1:800 |
| Antibody | CD95 (Hamster monoclonal) | BD Biosciences | 563646 | FCM 1:800 |
| Antibody | CXCR5 (Rat monoclonal) | BD Biosciences | 551961 | FCM 1:100 |
| Antibody | GL7 (Rat monoclonal) | BD Biosciences | 562967 | FCM 1:400 |
| Antibody | F4/80 (Rat monoclonal) | eBioscience | 25-4801-82 | FCM 1:400 |

*Continued on next page*

*Continued*

| Reagent type (species) or resource | Designation | Source or reference | Identifiers | Additional information |
|---|---|---|---|---|
| Antibody | Ly6C (Rat monoclonal) | eBioscience | 45-5932-82 | FCM 1:800 |
| Antibody | MHC II (I-A/I-E) (Mouse monoclonal) | eBioscience | 56-5321-82 | FCM 1:800 |
| Antibody | CD209b (SIGN-R1) (Hamster monoclonal) | eBioscience | 16-2093-82 | IF 1:200 |
| Antibody | Anti-Rabbit IgG (Goat polyclonal) | Life Technologies | A-11070 | FCM 1:1000 |
| Antibody | Granzyme B (Mouse monoclonal) | Invitrogen | GRB05 | FCM 5 µl |
| Antibody | p-RIP3 (T231/S232) (Rabbit monoclonal) | Cell Signaling Technology | 91702 | IF 1:200 |
| Antibody | p-MLKL (S345) (Rabbit monoclonal) | Cell Signaling Technology | 37333 | IF 1:200 |
| Antibody | Cleaved caspase-3 (Rabbit polyclonal) | Cell Signaling Technology | 9661 | IF 1:400 |
| Antibody | MLKL (Rabbit monoclonal) | Cell Signaling Technology | 37705 | WB 1:1000 |
| Antibody | Cleaved caspase-1 (Rabbit polyclonal) | Santa Cruz Biotechnology | sc-514 | WB 1:500 |
| Antibody | Beta-actin (Rabbit polyclonal) | Cell Signaling Technology | 4967 | WB 1:1000 |
| Antibody | p-MLKL (S345) (Rabbit monoclonal) | Abcam | ab196436 | WB 1:1000 |
| Antibody | mouse IgG1-HRP (Goat polyclonal) | Southern Biotech | 1070–05 | ELISA 1:5000 |
| Antibody | mouse IgG2b-HRP (Goat polyclonal) | Southern Biotech | 1090–05 | ELISA 1:5000 |
| Antibody | mouse IgG2c-HRP (Goat polyclonal) | Southern Biotech | 1079–05 | ELISA 1:5000 |
| Peptide, recombinant protein | K(b)/Ova. SIINFEKL tetramer | NIH Tetramer Core Facility | | FCM 1:400 |

*Continued on next page*

*Continued*

| Reagent type (species) or resource | Designation | Source or reference | Identifiers | Additional information |
|---|---|---|---|---|
| Peptide, recombinant protein | EndoGrade Ovalbumin | Hyglos | 321001 | For injection |
| Peptide, recombinant protein | Ova-AF647 | Life Technologies | O34784 | |
| Peptide, recombinant protein | Albumin, chicken egg white (Ovalbumin) | Sigma-Aldrich | A2512 | For ELISA |
| Peptide, recombinant protein | Uricase from Candida sp. | Sigma-Aldrich | U0880-250UN | |
| Peptide, recombinant protein | DNase I | Roche | 10104159001 | |
| Peptide, recombinant protein | Murine GM-CSF | PeproTech | 315–03 | |
| Peptide, recombinant protein | Murine M-CSF | PeproTech | 315–02 | |
| Peptide, recombinant protein | Collagenase IV | Worthington Biochemical | LS004189 | |
| Peptide, recombinant protein | Streptavidin-PE | BD Biosciences | 554061 | |
| Chemical compound, drug | OptEIA TMB Substrate | BD Biosciences | 555214 | |
| Peptide, recombinant protein | Annexin V | Biolegend | 640906 | FCM 5 µl |
| Chemical compound, drug | MF59 | Novartis | | |
| Chemical compound, drug | AddaVax | InvivoGen | vac-adx-10 | |
| Chemical compound, drug | Alhydogel | InvivoGen | vac-alu-250 | |
| Chemical compound, drug | z-VAD-fmk | Cayman Chemical | 14463 | |
| Chemical compound, drug | Nec-1s | BioVision | 2263–5 | |
| Chemical compound, drug | Live/Dead Aqua stain kit | Invitrogen | L34957 | FCM 1:1000 |
| Chemical compound, drug | Blotting-grade blocker | Bio-Rad | 1706404 | |

*Continued on next page*

*Continued*

| Reagent type (species) or resource | Designation | Source or reference | Identifiers | Additional information |
|---|---|---|---|---|
| Chemical compound, drug | Protease/ phosphatase inhibitor cocktail | Cell Signaling Technology | 5872 | |
| Chemical compound, drug | SuperSignal West Femto/Dura chemiluminescent substrate | Thermo Scientific | 34094/34075 | |
| Commercial assay or kit | Mouse IL-1b ELISA kit | BD Biosciences | 559603 | |
| Commercial assay or kit | Mouse IL-6 ELISA kit | BD Biosciences | 555240 | |
| Commercial assay or kit | Mouse IL-12 (p70) ELISA Set | BD Biosciences | 555256 | |
| Commercial assay or kit | Mouse TNF ELISA Set II | BD Biosciences | 558534 | |
| Commercial assay or kit | Quanti-iT PicoGreen dsDNA assay kit | Invitrogen | P11496 | |
| Commercial assay or kit | Uric Acid Assay kit | Abcam | ab65344 | |
| Commercial assay or kit | Standard macrophage depletion kit | Encapsula NanoSciences | 8901 | Clodronate liposome |
| Software, algorithm | FlowJo | BD | | |
| Software, algorithm | Prism | GraphPad | | |
| Software, algorithm | CellProfiler | Broad Institute | | |

## Mice and immunization

C57BL/6, Batf3 KO and Caspase 1 KO mice were purchased from Jackson Laboratories. RIPK3 KO (*Ripk3*$^{-/-}$) (*Newton et al., 2004*) and RIPK3 kinase-inactive (*Ripk3*$^{K51A/K51A}$) (*Mandal et al., 2014*) were kindly provided by E. Mocarski (Emory University). Tissues from immunized CD169-DTR and LysM-iDTR mice were kindly provided by M. Merad (Icahn School of Medicine at Mount Sinai). ASC KO mice and MLKL KO mice were originally obtained from V. M. Dixit (Genentech) and bred under specific pathogen-free conditions. Mice were immunized subcutaneously at the base of tail with endotoxin-free Ovalbumin (Hyglos) and following adjuvants: MF59 (Novartis), Addavax (Invivogen) and Alum (Alhydrogel; Invivogen). In case of co-administration of cell death inhibitors such as z-VAD-fmk (500 µg/mouse) and Nec-1s (100 µg/mouse) or enzymes including DNaseI (2,000U/mouse), those reagents were mixed with vaccine inoculums right before the subcutaneous injections. In order to deplete LN-resident macrophages, 0.25 mg (50 µl) of Clodrosome (CLL) and Encapsome (control liposome) were subcutaneously injected 5 days prior to each vaccination. Mice were maintained under specific-pathogen-free conditions in the vivarium of Emory Vaccine Center. All animal studies were conducted by following animal protocols reviewed and approved by the Institutional Animal Care and Use Committee of Emory University.

## Antibodies and flow cytometry

Cell suspensions from PBMC, dLN, spleen, lung, liver and small intestine were stained with following fluorochrome-labeled antibodies. The following antibodies were obtained from BioLegend: CD11c (N418), CD169 (SER-4), CD19 (6D5), CD3 (17A2), CD317 (PDCA-1; 927), CD4 (RM4-5), CD44 (IM7), B220 (RA3-6B2), CD80 (16-10A1), CD86 (GL1), CD8 (53–6.7), IFN-g (XMG1.2), IL-5 (TRFK5), Ly6G

(Gr-1; 1A8), TCR-b (H57-597), XCR1 (ZET). The following antibodies were obtained from BD Biosciences: CD103 (M290), CD11b (M1/70), CD279 (PD-1; J43), CD45 (30-F11), CD62L (MEL-14), CD95 (Fas; Jo2), CXCR5 (2G8), GL7 (GL7), KLRG1 (2F1). The following antibodies were obtained from eBioscience: F4/80 (BM8), Ly6C (HK1.4), MHC II (I-A/I-E; M5/114.15.2), pro-IL-1b (NJTEN3), CD209b (SIGN-R1; 22D1). Other reagents were Annexin V and 7-AAD (Biolegend), anti-Rabbit IgG and AF647-conjugated Ova (Life Technologies), granzyme B (GB11; invitrogen) and PE-conjugated streptavidin (BD). The following reagent was obtained through the NIH Tetramer Core Facility: K(b)/Ova. SIINFEKL.

To distinguish live/dead cells, cells were stained with Aqua (Thermo) or ef780 (eBioscience) prior to staining with antibodies. Fc Block (BD Biosciences) was added. For intracellular staining, cells were incubated in 100 ul of Lyse/Fix buffer (BD) and antibodies were added in 1x Cytofix/perm buffer (BD). Samples were acquired on a BD LSR II or BD LSRFortessa, and data were analyzed using FlowJo.

## Antibody ELISA

Nunc Maxisorp 96-well plates were coated with 10 μg/mL Ovalbumin (grade VI, SIGMA-ALDRICH), goat anti- mouse IgG (1 μg/mL) (Southern Biotech) and incubated overnight at 4°C. Plates were washed 3 times with 0.5% Tween-20 in PBS and blocked with 200 μl of 2% blotting-grade blocker (Biorad). Mouse serum samples were serially diluted in blocking buffer and incubated on blocked plates. Antigen-specific serum antibodies were detected using horseradish peroxidase (HRP)-conjugated antibodies (anti-mouse IgG, anti-mouse IgG1, anti-mouse IgG2b, anti-mouse IgG2c, and anti-mouse IgE) (Southern Biotech) at 1:5000 dilution in blocking buffer. Incubation of serum samples or antibodies was conducted at room temperature. HRP activity was detected using 100 μL of tetramethylbenzidine (TMB) substrate (BD Biosciences) and stopped using 50 μL 2N $H_2SO_4$. Developed plates were recorded using BioRad spectrophotometer at 450 nm with correction at 595 nm by subtraction.

## Cytokine ELISA and DAMPs detection

Serum samples were collected from immunized mice. Cytokine levels were determined in serum samples by sandwich ELISA. IL-1β, IL-6, TNFα and IL-12p70 were measured using OptEIA ELISA sets (BD Biosciences). ELISAs were performed according to manufacturer's instructions. Serum levels of dsDNA were determined by Quant-iT PicoGreen dsDNA Kit (invitrogen). Uric acid amount in serum was measured by Uric Acid Assay Kit (abcam).

## Immunohistochemistry and confocal microscopy

Isolated inguinal LNs were fixed for 2 hr in 4% paraformaldehyde (PFA) and were saturated in 30% sucrose overnight. LNs were embedded in optimal cutting temperature (OCT) medium, then were frozen and cut into 10–20 μm thick sections. Sections were stained with fluorochrome-conjugated antibodies (SIGN-R1, CD11b, B220, CD169 and cleaved caspase 3) and were imaged on a Leica SP8 confocal microscope with integrated 'tunable' filter sets under identical conditions in the same imaging session. Fluorophores were selected such that each fluorophore is tied to a dedicated laser to limit crossover excitation. Sequential scanning and image-recompiling were used to reduce fluorophore bleed through, and filter sets were set not to exceed 40 nm to ensure tight control over independent fluorophore signal capture. Laser power was set independently for each fluorophore to identify potential oversaturation (preventing reliable image quantitation) or bleed-through issues into other channels. Signals deemed at-risk for bleed through were quantified to ensure signal independence. phospho-RIPK3, phospho-MLKL, Caspase 3 signal (AF488) and SIGNR-1 (AF568) were assessed using total fluorescence measurements across the image and shown to be independent. Acquired images were analyzed with CellProfiler software (Broad Institute) for the identification of individual macrophages in each section.

## Electron microscopy

Mouse lymph node samples were fixed with 2.5% glutaraldehyde in 0.1M cacodylate buffer (pH 7.4). Samples were then washed and post-fixed with 1% osmium tetroxide in the same buffer for 1 hr. After rinsing with deionized water, samples were dehydrated through an ethanol series and then

placed in 100% ethanol. Following dehydration, lymph node samples were placed in propylene oxide for 10 min before infiltration with propylene oxide and Eponate 12 resin (Ted Pella, Inc, Redding, CA) at a 1:1 ratio overnight. After additional infiltration in pure Epnonate 12 resin, lymph node samples were placed in labeled Beem capsule and polymerized in a 60°C oven. Ultrathin sections were cut at 70–80 nm thick on a Leica UltraCut S ultramicrotome (Leica Microsystems Inc, Buffalo Grove, IL). Grids with ultrathin sections were stained with 5% uranyl acetate and 2% lead citrate. Ultrathin sections were imaged on a JEOL JEM-1400 transmission electron microscope (JEOL Ltd., Tokyo, Japan) equipped with a Gatan US1000 CCD camera (Gatan, Pleasanton, CA).

## Antigen Cross-presentation assay

For bone marrow-derived macrophages (BMMs) and DCs (BMDCs), tibiae and femurs were harvested from C57BL/6 mice or RIPK3 KO mice. Bones were washed in 70% ethanol and flushed with ice-cold HBSS through a 70 µm cell strainer. After red blood cell lysis, BM cells were pelleted and plated at a density of 5–7 $\times 10^6$ bone marrow cells per 10 cm Petri dish in (RPMI complete) in the presence of M-CSF (20 ng/ml, Peprotech) or GM-CSF (20 ng/ml, Peprotech). At day 5, the BMM culture was supplemented with fresh media with M-CSF, and harvested by StemPro Accutase (Life Technologies) treatment and gentle flushing at day 7. For the BMDCs, media was replaced with fresh media supplemented with GM-CSF at day 4 and 6, and cells were harvested in the same way with BMM at day 8. BMMs were seeded in 24 well plate (tissue culture non-treated), and were cultured with different concentration of AV with Ova for indicated times. BMM cells were washed, collected and added to BMDC culture (round-bottom 96-well plate, tissue culture non-treated) for 20 hr. After that, BMDCs were thoroughly washed with PBS, and added with CFSE-labeled naïve OT-I cells. 3 days later, the proliferation of OT-I cells was assessed by flow cytometry.

## Western blot

LNs were collected in ice-cold RIPA lysis buffer supplemented with 1x protease/phosphatase inhibitor cocktail (Cell Signaling Technology). After snap freeze and thaw by liquid nitrogen, LNs were homogenized via BioMasher II (RIP Research Products), followed by sonication. Equal amounts of protein from whole LN lysates were run on an SDS-PAGE and transferred onto nitrocellulose membranes. After blocking with 5% fat-free milk, the membranes were incubated at 4°C with the following primary antibodies: anti-mouse p-MLKL (S345; ab196436, Abcam), MLKL (28640, Cell Signaling Technology), cleaved caspase 1 (M-20, Santa Cruz), cleaved caspase 3 (9661, Cell Signaling Technology) and β-actin. The membranes were then washed and incubated with Horseradish peroxidase–conjugated secondary antibody (Cell Signaling Technology). Proteins were visualized with SuperSignal West Femto or Dura chemiluminescent substrate (Pierce). Signals were acquired and analyzed via Odyssey Fc (LI-COR).

## Statistical analysis

All results are displayed as mean ± s.e.m. Biological replicates were used in all experiments unless stated otherwise. Statistical significance was determined by t-test, one- way ANOVA, or two-way ANOVA using Prism software (GraphPad) depending on the experimental layout. Survival analysis was determined by Log-rank (Mantel-Cox) test. Probability values of $p < 0.05$ were considered significant and denoted by *. Where indicated, ** denotes $p < 0.01$, ***$p < 0.001$ and ***$p < 0.0001$.

## Acknowledgements

We thank Derek O'Hagan at GSK for providing us with MF59. We are grateful to Meera Trisal for her technical help with the MLKL knockout experiments. We acknowledge the NIH (grants R37 DK057665, R37 AI048638, U19 AI090023, and U19 AI057266) and the Bill and Melinda Gates Foundation, and the Soffer Fund endowment for supporting this work in Bali Pulendran's lab. In addition, this study was supported in part by the Robert P Apkarian Integrated Electron Microscopy Core (RPAIEMC), which is subsidized by the Emory College of Arts and Sciences and the Emory University School of Medicine and is one of the Emory Integrated Core Facilities. Additional support was provided by the National Center for Advancing Translational Sciences of the National Institutes of Health under award number UL1TR000454. The content is solely the responsibility of the authors and does not necessarily reflect the official views of the National Institutes of Health. The data

described here were gathered on the JEOL JEM-1400 120kV TEM supported by a National Institutes of Health Grant S10 RR025679.

## Additional information

### Funding

| Funder | Grant reference number | Author |
|---|---|---|
| National Institutes of Health | R37 DK057665 | Bali Pulendran |
| National Institutes of Health | R37 AI048638 | Bali Pulendran |
| National Institutes of Health | U19 AI057266 | Bali Pulendran |
| National Institutes of Health | U19 AI090023 | Bali Pulendran |
| Bill and Melinda Gates Foundation | | Bali Pulendran |
| Soffer Fund | | Bali Pulendran |

The funders had no role in study design, data collection and interpretation, or the decision to submit the work for publication.

### Author contributions

Eui Ho Kim, Data curation, Formal analysis, Validation, Investigation, Visualization, Methodology, Writing; Matthew C Woodruff, Data curation, Investigation, Methodology, Writing - review and editing; Lilit Grigoryan, Investigation; Barbara Maier, Resources, Investigation, Methodology; Song Hee Lee, Rajesh Ravindran, Huailiang Ma, Resources, Investigation; Pratyusha Mandal, Alexander D Gitlin, Investigation, Writing - review and editing; Mario Cortese, Muktha S Natrajan, Resources, Project administration; Miriam Merad, Resources; Edward S Mocarski, Conceptualization, Resources; Joshy Jacob, Resources, Supervision; Bali Pulendran, Conceptualization, Supervision, Funding acquisition, Project administration

### Author ORCIDs

Eui Ho Kim (iD) https://orcid.org/0000-0002-4235-5622
Matthew C Woodruff (iD) https://orcid.org/0000-0002-5252-7539
Bali Pulendran (iD) https://orcid.org/0000-0001-6517-4333

### Ethics

Animal experimentation: This study was performed in strict accordance with the recommendations in the Guide for the Care and Use of Laboratory Animals of the National Institutes of Health. All of the animals were handled according to approved institutional animal care and use committee (IACUC) protocols (#2002593) of Emory University.

### Decision letter and Author response

Decision letter https://doi.org/10.7554/eLife.52687.sa1
Author response https://doi.org/10.7554/eLife.52687.sa2

## Additional files

### Supplementary files

• Transparent reporting form

### Data availability

All data generated or analysed during this study are included in the manuscript and supporting files.

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
