## [Decision Letter]

Thank you for submitting your article "Squalene-based adjuvants stimulate CD8 T-cell responses, but not antibody responses, through a RIP3k necroptotic pathway" for consideration by *eLife*. Your article has been reviewed by two peer reviewers, and the evaluation has been overseen by a Reviewing Editor and Tadatsugu Taniguchi as the Senior Editor. The following individuals involved in review of your submission have agreed to reveal their identity: Andrew Oberst (Reviewer #1); Ross Kedl (Reviewer #2).

The reviewers have discussed the reviews with one another and the Reviewing Editor has drafted this decision to help you prepare a revised submission.

Summary:

This manuscript presents a role for the necroptotic death of lymph node resident macrophages as a key mechanism by which squalene-based adjuvants elicit CD8^+^ T cell immunity. The authors walk their way through a well-designed series of experiments to support the conclusions that: 1) squalene-based adjuvants induce both necrosis and apoptosis in the draining lymph node macrophages, 2) necroptotic medullary sinus marophages are picked up by Batf3^+^ dendritic cells and antigen is cross presented to CD8^+^ T cells, and 3) these CD8^+^ T cell responses are RIP3K-dependent whereas B cell responses are unaffected by necroptosis, instead relying on apoptosis of macrophages to augment antibody responses. The conclusions, which are well supported by the data, will be of high interest to the biological community.

Essential revisions:

Below are compiled comments from the reviewers, each one numbered by general topic. The reviewers have requested that additional experiments be performed to address #1-2. If possible, experiments to address #3 are highly desirable, unless the authors have sufficient justification to decide otherwise.

1) Studies in MLKL knockouts

Reviewer 1: An emerging paradigm in RIPK3 signaling is that cell death is not the only, or even most important, immunological output of this pathway. The paper by Yatim et al. cited by the authors actually indicates that it is RIPK1- and NF-ΚB-dependent cytokines produced following RIPK3 activation that underlies the observed effect. Further evidence is provided by K. Newton et al. Cell Death Differ 2016. Given this, it would be of great interest to clarify whether the observed effect actually requires necroptotic cell death and "DAMP" release at all. The RIPK3 K51A mice don't really address this, as they lack both the death and the cytokine outputs of RIPK signaling. The simplest way to do this would be to run the vaccination model in mice lacking MLKL, the effector of lytic necroptosis. Alternatives could include using the in vitro co-culture model described by the authors in combination with NF-κB and/or MLKL inhibition. Assaying direct de novo production of inflammatory cytokines from WT or RIPK3 KO LN macrophages upon AV treatment would also be illuminating.

Reviewer 2: An additional experiment is the MLKL KO response, as MLKL is the terminal effector molecule of the necroptotic pathway… if the predictions are correct, that should have poor CD8 responses but OK Ab.

2) More direct demonstration of necroptosis in vivo.

Reviewer 1: I find the IF data presented in Figure 4D difficult to understand. The previous figures show an accumulation of cleaved Casp3 at late timepoints only, which is preceded by MLKL phosphorylation. However, this figure appress to show very early caspase-3 cleavage upon AV immunization. More to the point, robust antibodies exist to phosphorylated RIPK3 and MLKL, which work well in IF and IHC. Use of these reagents, and demonstration of in vivo necroptosis induced by AV, would much more clearly make the authors' point.

3) An in vivo readout to demonstrate the importance of RIPK3 in AddaVax-induced CD8 T cell responses.

Reviewer 1: While the assays used point to a requirement for RIPK3 in the emergence of CD8^+^ T cell immunity upon AV vaccination, the paper currently lacks an in vivo readout that this difference is important. A simple experiment using the systems the authors already have in place would be to repeat the B16 tumor clearance experiment in RIPK3 knockout mice. Showing that alum-based vaccination does not protect mice from this model would likewise strengthen the conclusions.

[Editors' note: further revisions were suggested prior to acceptance, as described below.]

Thank you for submitting your article "Squalene-based adjuvants stimulate CD8 T cell, but not antibody responses, through a RIPK3-dependent pathway" for consideration by *eLife*. Your article has been reviewed by two peer reviewers, and the evaluation has been overseen by a Reviewing Editor and Tadatsugu Taniguchi as the Senior Editor.

The reviewers have discussed the reviews with one another and the Reviewing Editor has drafted this decision to help you prepare a revised submission.

Summary:

In this revised manuscript, the authors have addressed most of the specific points I raised, and there is consensus that requesting additional work or major modifications at this stage would be both unfair and unnecessary.

Revisions:

Some remaining issues that would benefit from adding to the text/Discussion, or where appropriate, providing a response to reviewers are:

1) The uric acid release and the role for RIPK3 in CD8^+^ T cell responses may very well be true, true and unrelated. I am not convinced it is sufficiently grounded in causation to include. Especially since it is also highly released in response to Alum, it cannot contribute to understanding the specific mechanism of the AV. This fact is not addressed in the uricase treatment in Figure 6 because it is only AV immunization and no alum is included. As both alum and AV induce copious amounts of uric acid, were uricase treatment to have an impact on AVC but not alum immunized mice, this would nicely serve to implicate it in the AV-based mechanism of CD8 induction. Otherwise we are back to true true and unrelated… at least unrelated to mechanisms that delineate AV from alum.

2) Chlodronate depletes both macrophages and splenic cDC2s. (Ciavarra et al. 2005). This is worth noting. It is not clear to me what impact the various DTR mice (other than the BatF3) have on cDC2s, but it seems at least worth an acknowledgement that they may have a role that may not be able to be ruled out by the given experiments.

3) The CFSE data (6E) are still problematic if only percentages are going to be shown and not total numbers. In their response to the review, the authors stated that the number of undivided cells are different between wt and KO BMMs. However, only percentages are shown, a result totally consistent with identical numbers of undivided cells (which I expect it probably true) but differential accumulation of cells that are divided. If the data are going to be kept then total numbers need to be added and the exact impact of their KO on CD8 division clarified.

4) Again, uricase treatment in the last figure is of highly questionable value to the specific mechanism of AV as compared to alum given that both induce copious production of uric acid (Figure 4H). Did the authors do alum side by side here?… if there is a difference in the AV response, but not the alum response, after uricase treatment, then please show it. If not, then we can all rule out Uric acid as the DAMP that is specific to the AV-RIPK axis of T cell induction. This does not compromise the fact that there appears to be a role for RIPK in Cd8 responses that was previously unidentified.

5) As their MLKL data indicate that actual death by necroptosis has no impact on the immune responses, doesn't their model at the end need modification? Shouldn't it rely more on the signaling of RIPK, in either T cells or APC, than any significant role for necroptotic cell death? Wasn't that the point of the MLKL mice? Similarly, is the nec-1 data in the last figure necessary? The MLKL data are far more encompassing… is there no antibody data form the MLKL immunizations? If not I guess this is a decent proxy, but then it should be stated that this is unsurprising, given the lack of any effect in the MLKL KO. Lastly, it seems to me that the Discussion should be modified to temper the conclusion that over necroptosis is a critical component of their RIPK-dependent mechanism.

---

## [Author Response]

Essential revisions:Below are compiled comments from the reviewers, each one numbered by general topic. The reviewers have requested that additional experiments be performed to address #1-2. If possible, experiments to address #3 are highly desirable, unless the authors have sufficient justification to decide otherwise.1) Studies in MLKL knockoutsReviewer 1: An emerging paradigm in RIPK3 signaling is that cell death is not the only, or even most important, immunological output of this pathway. The paper by Yatim et al. cited by the authors actually indicates that it is RIPK1- and NF-ΚB-dependent cytokines produced following RIPK3 activation that underlies the observed effect. Further evidence is provided by K. Newton et al. Cell Death Differ 2016. Given this, it would be of great interest to clarify whether the observed effect actually requires necroptotic cell death and "DAMP" release at all. The RIPK3 K51A mice don't really address this, as they lack both the death and the cytokine outputs of RIPK signaling. The simplest way to do this would be to run the vaccination model in mice lacking MLKL, the effector of lytic necroptosis. Alternatives could include using the in vitro co-culture model described by the authors in combination with NF-κB and/or MLKL inhibition. Assaying direct de novo production of inflammatory cytokines from WT or RIPK3 KO LN macrophages upon AV treatment would also be illuminating.Reviewer 2: An additional experiment is the MLKL KO response, as MLKL is the terminal effector molecule of the necroptotic pathway… if the predictions are correct, that should have poor CD8 responses but OK Ab.

The authors thank the reviewers for their shared perspective on the benefit of incorporating studies using the MLKL KO model into this manuscript. We agree that previous reports by Yatim et al. demonstrated that NF-ΚB-dependent cytokines (perhaps IL-6 or TNF) following RIPK3 activation induced CD8 T cell response in their experimental system. Therefore, the authors agree that the incorporation of a model targeting the downstream effector of necroptosis is of inherent interest to the manuscript regardless of the outcome. As a result, we have incorporated new vaccination data into the manuscript comparing the CD8^+^ T cell response in Wild type, vs MLKL KO vaccine recipients. The results of those data indicate that MLKL, an executioner of necroptosis, is not required for the SE adjuvant-induced CD8 T cell response (Figure 6D). This result is consistent with previous reports the reviewers mentioned, and the activation of RIPK3 signalling is critical for the CD8 T cell response possibly via secretion of inflammatory cytokines like IL-6 and TNF.

Meanwhile, the authors think that a certain DAMP is required for this response since in vivo blocking uric acid by uricase treatment significantly impaired the AV-induced CD8 T cell response. Mulay et al. has reported that MSU can induce RIPK3-dependent necroptosis, so it is likely that SE adjuvant-derived uric acid stimulates RIPK3 signalling, followed by the optimal induction of CD8 T cell response.

2) More direct demonstration of necroptosis in vivo.Reviewer 1: I find the IF data presented in Figure 4D difficult to understand. The previous figures show an accumulation of cleaved Casp3 at late timepoints only, which is preceded by MLKL phosphorylation. However, this figure appress to show very early caspase-3 cleavage upon AV immunization. More to the point, robust antibodies exist to phosphorylated RIPK3 and MLKL, which work well in IF and IHC. Use of these reagents, and demonstration of in vivo necroptosis induced by AV, would much more clearly make the authors' point.

The authors appreciate the perspective that directly staining mediators of necroptosis would be a more direct test of necroptosis pathway execution at the time points indicated by other figures in the paper. Indeed, the use of the cleaved caspase-3 IF data implying cell death signalling at later time points, while supporting the idea of ongoing death in the lymph node, was ultimately out of step with the core messaging of the manuscript.

To address these concerns, we have replaced the cleaved caspase IF data with a more relevant time course showing the robust phosporylation of both RIP3K and MLKL by 6 hrs post vaccination and persisting to at least 24h (Figure 4C). These new data have replaced Figure 4D, with an additional panel included to show the lack of MLKL phosphorylation in RIPK3 KO controls to help validate the specificity of the staining.

It should be pointed out that while the necroptosis staining is identified (as expected) in the medulla of the draining inguinal lymph nodes following vaccination in line with the extensively described loss of medullary macrophages in this manuscript, there was an unexpected loss of SIGNR-1 surface staining on pMLKL and pRIPK3 ^+^ cells. To address this issue, we have incorporated additional staining parameters in an attempt to accurately identify the necroptotic cells as medullary macrophages. Through the use of CD11b, F4/80, MHCII, and CD169, we have.

3) An in vivo readout to demonstrate the importance of RIPK3 in AddaVax-induced CD8 T cell responses.Reviewer 1: While the assays used point to a requirement for RIPK3 in the emergence of CD8^+^ T cell immunity upon AV vaccination, the paper currently lacks an in vivo readout that this difference is important. A simple experiment using the systems the authors already have in place would be to repeat the B16 tumor clearance experiment in RIPK3 knockout mice. Showing that alum-based vaccination does not protect mice from this model would likewise strengthen the conclusions.

The authors agree that the suggested experiments may strengthen the RIPK3-dependent mechanism for SE-induced CD8 T cell response. Actually, several additional experiments were carried out to verify if the adjuvant-induced CD8 T cell response was essential for tumor rejection. For example, Ova+alum vaccination also provided partial protection (which was weaker than Ova+AV group) to the tumor challenge possibly due to Ova-specific antibody derived ADCC effect (Data not shown). In addition, depletion of CD8 T cells by rat IgG (53-6.7) followed by Ova+AV vaccination failed to generate normal protection after the tumor challenge (Data not shown).

[Editors' note: further revisions were suggested prior to acceptance, as described below.]

Revisions:Some remaining issues that would benefit from adding to the text/Discussion, or where appropriate, providing a response to reviewers are:1) The uric acid release and the role for RIPK3 in CD8^+^ T cell responses may very well be true, true and unrelated. I am not convinced it is sufficiently grounded in causation to include. Especially since it is also highly released in response to Alum, it cannot contribute to understanding the specific mechanism of the AV. This fact is not addressed in the uricase treatment in Figure 6 because it is only AV immunization and no alum is included. As both alum and AV induce copious amounts of uric acid, were uricase treatment to have an impact on AVC but not alum immunized mice, this would nicely serve to implicate it in the AV-based mechanism of CD8 induction. Otherwise we are back to true true and unrelated… at least unrelated to mechanisms that delineate AV from alum.

After combing through the presented data, the authors still believe that uric acid likely plays a role in the induction of the CD8 T cell response in the draining lymph nodes following AV administration. However, they also understand the reviewers concerns about the specificity of uric acid in the differentiation between alum and AV responses as administration of both leads to detectible uric acid release in the serum. We suspect that uric acid may serve as a general driver of responses that are polarized by the different lymph node microenvironments resulting from alum versus AV administration. We have not, however, described those other putative environmental factors in detail sufficiently to warrant inclusion within this manuscript. Therefore, In line with the reviewer’s recommendation, we have decided to remove our descriptions of uric acid as a specific factor in driving the RIPK3-dependent signaling pathway and have substantially altered both the manuscript and the figures to that effect.

2) Chlodronate depletes both macrophages and splenic cDC2s. (Ciavarra et al. 2005). This is worth noting. It is not clear to me what impact the various DTR mice (other than the BatF3) have on cDC2s, but it seems at least worth an acknowledgement that they may have a role that may not be able to be ruled out by the given experiments.

The reviewer brings up an interesting point about the new characterization of cDC2s, and the authors agree with the caveat of macrophage depletion using chlodronate-loaded liposome (CLL) method. Here, we first used CLL to generally assess the role of mostly macrophages (and possibly some other phagocytes such as monocytes and CD8- DCs). After seeing impaired phenotype in CD8 T cell response, we wanted to more specifically characterize the role of macrophage subsets in immune response using LysM-iDTR and CD169-DTR. Together, we believe that our model still strongly points to macrophages as the likely target of this pathway, but we cannot fully rule out other population’s contributions as we have not directly tested their status across our various knockout models. As a result, we have modified the text to leave open this possibility.

3) The CFSE data (6E) are still problematic if only percentages are going to be shown and not total numbers. In their response to the review, the authors stated that the number of undivided cells are different between wt and KO BMMs. However, only percentages are shown, a result totally consistent with identical numbers of undivided cells (which I expect it probably true) but differential accumulation of cells that are divided. If the data are going to be kept then total numbers need to be added and the exact impact of their KO on CD8 division clarified.

The authors appreciate the reviewer’s point on the CFSE experiment presented in Figure 6E. Looking back and reanalyzing the original data, we realized that the max values of Y-axis of histograms were normalized, and that’s why the numbers of undivided cells (max peaks) look identical. In Author response image 1 is histograms using Y axis with actual cell counts (and normalized Y-axis) and numbers of undividing and dividing cells of each sample. In addition, we have updated the Figure 6E with the corrected histograms.

4) Again, uricase treatment in the last figure is of highly questionable value to the specific mechanism of AV as compared to alum given that both induce copious production of uric acid (Figure 4H). Did the authors do alum side by side here?… if there is a difference in the AV response, but not the alum response, after uricase treatment, then please show it. If not, then we can all rule out Uric acid as the DAMP that is specific to the AV-RIPK axis of T cell induction. This does not compromise the fact that there appears to be a role for RIPK in Cd8 responses that was previously unidentified.

The authors collectively responded to the comments 1 and 4. Please see our response for 1.

5) As their MLKL data indicate that actual death by necroptosis has no impact on the immune responses, doesn't their model at the end need modification? Shouldn't it rely more on the signaling of RIPK, in either T cells or APC, than any significant role for necroptotic cell death? Wasn't that the point of the MLKL mice? Similarly, is the nec-1 data in the last figure necessary? The MLKL data are far more encompassing… is there no antibody data form the MLKL immunizations? If not I guess this is a decent proxy, but then it should be stated that this is unsurprising, given the lack of any effect in the MLKL KO. Lastly, it seems to me that the Discussion should be modified to temper the conclusion that over necroptosis is a critical component of their RIPK-dependent mechanism.

The authors appreciate the reviewer’s concern about the working model. We have revised the working model figure (Figure 7E) like below and also modified sentences in the Discussion to carefully describe what this study has revealed, especially emphasizing the role of RIPK3 signaling rather than “necroptosis” itself in the generation of CD8 T cell response by Ag+SE vaccination.

Regarding the antibody data, we did not measure Ag-specific antibody response in MLKL KO mice after the Ova+AV vaccination. Since IgG levels is normal in RIPK3 KO mice, we assume that the antibody response in MLKL KO mice would also be normal.